# FP-IRL: Fokker-Planck-based Inverse Reinforcement Learning — A Physics-Constrained Approach to Markov Decision Processes

## Abstract

Inverse Reinforcement Learning (IRL) is a compelling technique for revealing the rationale underlying the behavior of autonomous agents. IRL seeks to estimate the unknown reward function of a Markov decision process (MDP) from observed agent trajectories. While most IRL approaches require the transition function to be prescribed or learned a-priori, we present a new IRL method targeting the class of MDPs that follow the Itô dynamics without this requirement. Instead, the transition is inferred in a physics-constrained manner simultaneously with the reward functions from observed trajectories leveraging the mean-field theory described by the Fokker-Planck (FP) equation. We conjecture an isomorphism between the time-discrete FP and MDP that extends beyond the minimization of free energy (in FP) and maximization of the reward (in MDP). This isomorphism allows us to infer the potential function in FP using variational system identification, which consequently allows the evaluation of reward, transition, and policy by leveraging the conjecture. We demonstrate the effectiveness of FP-IRL by applying it to synthetic benchmarks and a biological problem of cancer cell dynamics, where the transition function is unknown.

## 1 Introduction

Principles may be unavailable for deciphering the incentive mechanism in a complex decision-making system, especially when we have poor or even no knowledge about the system (e.g., on the environment, agents, etc.). Important examples of this type arise in cancer biology where the mechanisms of cancer cell metastasis remain to be understood, and in human interactions where human agents may change unpredictably and into regimes not encountered previously. The stochasticity of the system and the heterogeneity among individuals (cells or humans) further complicate the problem. Nonetheless, learning incentives holds great potential for understanding these complex systems and eventually developing targeted interventions to control them. Inverse Reinforcement Learning (IRL) (Russell, 1998; Ng and Russell, 2000; Ratliff et al., 2006; Ramachandran and Amir, 2007; Ziebart et al., 2008; Fu et al., 2018) is a powerful tool that can aid in the data-driven recovery of incentive mechanisms that force the behavior of the target agent.

IRL has demonstrated remarkable success in diverse fields, such as human behaviors (Ratliff et al., 2006; Ziebart et al., 2008; Hossain et al., 2022), robotics (Levine and Koltun, 2012; Finn et al., 2016), and biology (Kalantari et al., 2020). However, it is not without limitations. Firstly, IRL typically requires access to sampling the next state from the environment through a prescribed or empirically estimated transition model. This can be problematic in situations where knowledge about the environment dynamics is lacking or imperfect, and accessibility of sampling from transition is not available. The examples of interactions between cancer cells or human agents also fall under this category. An empirical treatment of transition functions can be undesirable because it is often very challenging to generalize to state and action regions away from training samples relying on observations alone, especially under high-dimensional settings and when training data is limited and noisy. Secondly, recent IRL algorithms with unknown transitions may rely on purely data-driven deep learning techniques (Herman et al., 2016; Yue et al., 2023). However, the lack of interpretability in deep learning models can translate to difficulty in scientific understanding of the system behavior.

Many systems (e.g., swarms, crowd behavior) have mechanistic foundations, which if exploited can lead to better understanding and more efficient learning of their incentive structures. With the above motivation, we propose a new method of physics-constrained IRL. This method simultaneously estimates the transition and reward functions using only data on trajectories, while also inferring physical principles that govern the system and using them to constrain the learning. The key contributions of our work center around a conjecture on the structural isomorphism between the physics governed by a well-known optimal transport model—the Fokker-Planck (FP) equation—and Markov Decision Process (MDP). Using it, we leverage fundamental principles of the FP physics to build models for the MDP with computational benefits. We then exploit these theoretical and modeling insights and propose the physics-based FP-IRL algorithm. Finally, we demonstrate FP-IRL through numerical experiments on synthetic and real-world examples.

## 2 RELATED WORK

Studies most closely related to our work are as follows. Herman et al. (2016) introduced a purely data-driven IRL to simultaneously estimate the reward and transition using neural networks, but devoid of physics. Garg et al. (2021) proposed an IRL algorithm that learns the state-action value function first with a given transition and infers the reward function using the inverse Bellman operator. Lastly, Kalantari et al. (2020) applied a variant of Bayesian IRL to study gene mutations in cancer cell populations. In contrast to these, our approach will infer the transition and reward simultaneously but constrained by the FP physics, while exploiting the inverse Bellman operator in applications to study the migration dynamics of agents such as cancer cells. Besides these references, we briefly review other topics more broadly connected to our approach.

**Inverse Reinforcement Learning (IRL)** has the main goal of learning an unknown reward function (Russell, 1998; Ng and Russell, 2000). Many new IRL variants and extensions have since been developed. The maximum margin method (Ng and Russell, 2000; Ratliff et al., 2006) infers a reward function such that the expected reward of the demonstrated policy exceeds that of other sub-optimal policies by a maximal margin. The reward function inferred by the feature matching method (Abbeel and Ng, 2004) maximizes the margin while driving the resulting policy to be close to the demonstrated policy by comparing their feature counts. Entropy regularization has been added to feature matching to represent the uncertainty of predictions in (Ziebart et al., 2008; 2010; Ziebart, 2010). Generative imitation learning (Ho and Ermon, 2016) and adversarial IRL (Fu et al., 2018; Yu et al., 2019; Henderson et al., 2018) have extended entropy-regularized IRL to generative adversarial modeling. Offline IRL (Zeng et al., 2023; Yue et al., 2023) also learns a reward without the transition but it has to estimate the transition function by a data-driven approach prior to the inference of reward. Finally, Bayesian IRL (Ramachandran and Amir, 2007; Kalantari et al., 2020) computes the likelihood of trajectories given a reward function and uses Bayesian inference to quantify the uncertainty surrounding the reward function. Readers are directed to Arora and Doshi (2021) and Adams et al. (2022) for a complete survey on IRL.

**Entropy Regularized Reinforcement Learning (RL)** (also called soft RL or energy-based RL) uses the principle of maximum entropy to regularize reward inference (Ziebart et al., 2010) in order to obtain a robust optimal policy in an uncertain environment (Fox et al., 2016; Haarnoja et al., 2017; 2018a;b). The objective function bears a formal similarity to the free energy in statistical physics, but does not have the rigorous connection to it that we establish in this work.

**Free Energy Principle** proposes a general principle that defines a free energy related to information-theoretic ideas (Friston et al., 2006; Friston, 2009; 2010). When extended to RL (Friston et al., 2009), the information gain in this setting can be interpreted as the reward.

## 3 FOKKER-PLANCK-BASED INVERSE REINFORCEMENT LEARNING

In this section, we introduce the fundamentals of MDP and discuss the physics-based modeling of an MDP using FP in the IRL context. We then conjecture a structure isomorphism between FP and MDP, and propose a novel method to simultaneously estimate the transition, reward, and policy leveraging the conjecture. The overall FP-IRL method is summarized in Algorithm 1 in Appendix A.

## 3.1 PRELIMINARIES

A *Markov Decision Process (MDP)* is defined by a tuple $\mathcal{M} \triangleq \{\mathcal{S}, \mathcal{A}, p_0(\cdot), R(\cdot), T(\cdot)\}$ consisting of a state space $\mathcal{S} \subseteq \mathbb{R}^{d_s}$ with possible states $\boldsymbol{s} \in \mathcal{S}$, an action space $\mathcal{A} \subseteq \mathbb{R}^{d_a}$ with possible actions $\boldsymbol{a} \in \mathcal{A}$, initial state probability density function $p_0(\boldsymbol{s}) : \mathcal{S} \mapsto \mathbb{P}$, reward function $R(\boldsymbol{s}, \boldsymbol{a}) : \mathcal{S} \times \mathcal{A} \mapsto \mathbb{R}$ that evaluates the instantaneous scalar reward when taking action $\boldsymbol{a}$ at state $\boldsymbol{s}$, and transition probability function $T(\boldsymbol{s}'|\boldsymbol{s}, \boldsymbol{a}) : \mathcal{S} \times \mathcal{A} \times \mathcal{S} \mapsto \mathbb{P}$ that evaluates the probability of transitioning to state $\boldsymbol{s}'$ when taking action $\boldsymbol{a}$ at state $\boldsymbol{s}$.

In infinite-horizon MDP, *Reinforcement Learning (RL)* is concerned with finding an optimal time-invariant policy $\pi(\boldsymbol{a}|\boldsymbol{s}) : \mathcal{S} \times \mathcal{A} \mapsto \mathbb{P}$ (evaluating the probability of taking action $\boldsymbol{a}$ at state $\boldsymbol{s}$) that maximizes the expected cumulative discounted reward:

$$\pi^*(\cdot) = \underset{\pi(\cdot) \in \Pi}{\arg\max} \quad \underset{\substack{\mathbf{s}_0 \sim p_0(\cdot), \mathbf{a}_t \sim \pi(\cdot|\boldsymbol{s}_t), \\ \mathbf{s}_{t+1} \sim T(\cdot|\boldsymbol{s}_t, \boldsymbol{a}_t)}}{\mathbb{E}} \left[ \sum_{t=0}^{\infty} \gamma^t R(\boldsymbol{s}_t, \boldsymbol{a}_t) \right] \tag{1}$$

where $\gamma \in [0, 1)$ is the reward discount factor. The expected cumulative reward can be written in a recursive form, and the RL problem is equivalent to finding a policy maximizing the Bellman expectation equations (Bellman, 1952):

$$Q_\pi(\boldsymbol{s}, \boldsymbol{a}) = R(\boldsymbol{s}, \boldsymbol{a}) + \gamma \mathbb{E}_{\boldsymbol{s}' \sim T(\cdot|\boldsymbol{s}, \boldsymbol{a})} \left[ V_\pi(\boldsymbol{s}') \right], \tag{2a}$$

$$V_\pi(\boldsymbol{s}) = \mathbb{E}_{\mathbf{a} \sim \pi(\cdot|\boldsymbol{s})} \left[ Q_\pi(\boldsymbol{s}, \boldsymbol{a}) \right] \tag{2b}$$

where $Q(\boldsymbol{s}, \boldsymbol{a}) : \mathcal{S} \times \mathcal{A} \mapsto \mathbb{R}$ is the state-action value function that evaluates the expected cumulative rewards when choosing action $\boldsymbol{a}$ at state $\boldsymbol{s}$, and $V(\boldsymbol{s}) : \mathcal{S} \mapsto \mathbb{R}$ is the state value function that evaluates the expected cumulative rewards if the agent is at state $\boldsymbol{s}$.

*Inverse Reinforcement Learning (IRL)* is a problem where the goal is to infer *unknown* reward function $R(\cdot)$ from observed trajectories $\mathcal{D} \triangleq \left\{ (\boldsymbol{s}_0^{(i)}, \boldsymbol{a}_0^{(i)}, \cdots, \boldsymbol{s}_{\tau_i}^{(i)}, \boldsymbol{a}_{\tau_i}^{(i)}) \right\}_{i=1}^{m}$ ($m$ denotes the number of trajectories, $\tau_i$ the number of timesteps in the $i$-th trajectory) of a demonstrator (e.g., expert) who employs a policy that maximizes the unknown expected rewards. Conventionally, only reward $R(\cdot)$ is unknown from the MDP while all other components, including the transition function $T(\cdot)$, are assumed to be prescribed or empirically estimated prior to the reward inference. The transition is crucial to enable trajectory sampling, allowing the IRL problem to be tackled iteratively by adjusting the proposed $R(\cdot)$ so that the difference between simulated and observed trajectories is minimized.

In many real-life problems, transition function $T(\cdot)$ is also unknown (e.g., a probabilistic rule for cancer cell migration is not available) and not accessible for sampling when learning the reward. The absence of $T(\cdot)$ thus introduces indeterminacy, allowing many more transition-reward pairings to potentially describe the demonstrator behavior equally well. It is then crucial to combat this exacerbated ill-posedness by introducing additional regularization and constraints. Motivated by problems of cancer cell dynamics that are widely understood to be governed by different physical principles, we propose to achieve this by incorporating physical principles into IRL that will yield physically meaningful and interpretable results, instead of employing purely data-driven models for learning the transition and reward. The benefits of physics constraints in IRL are discussed in Sec. 5.

## 3.2 PHYSICS-BASED MODELING FOR LEARNING THE TRANSITION FUNCTION

The FP equation arises in many contexts in physics wherein the time evolution of a density function can be posed as an optimal transport map. It therefore provides a framework to model physical and biological systems of evolving distributions (Risken and Frank, 1996), and motivates our strategy to inject physics into IRL by constraining and learning the transition of the probability density function through the FP dynamics.

This is achieved by first recognizing that an MDP with a given policy $\pi(\cdot)$ reduces to an *Markov process (MP)* on the lumped state variable $\mathbf{x} = [\mathbf{s}, \mathbf{a}]$ ($\mathcal{X} \subseteq \mathbb{R}^d$) where the MP transition is

$$T_{\mathrm{MP}}(\boldsymbol{x}'|\boldsymbol{x}) = T_{\mathrm{MP}}(\boldsymbol{s}', \boldsymbol{a}'|\boldsymbol{s}, \boldsymbol{a}) = \pi(\boldsymbol{a}'|\boldsymbol{s}')T(\boldsymbol{s}'|\boldsymbol{s}, \boldsymbol{a}) \tag{3}$$

with $\pi(\boldsymbol{a}'|\boldsymbol{s}') = \pi(\boldsymbol{a}'|\boldsymbol{s}', \boldsymbol{s}, \boldsymbol{a})$ due to the Markov property. Then, inferring the MP transition enables the retrieval of the MDP transition via probability marginalization:

$$T(\boldsymbol{s}'|\boldsymbol{s}, \boldsymbol{a}) = \int_{\mathcal{A}} T_{\mathrm{MP}}(\boldsymbol{s}', \boldsymbol{a}'|\boldsymbol{s}, \boldsymbol{a}) \, d\boldsymbol{a}'. \tag{4}$$

Learning the MP transition will leverage connections between MPs and stochastic differential equations (SDEs). Specifically, we target the class of stochastic processes whose dynamics are governed by the Itô SDE (e.g., FP dynamics, many real-world problems including cell dynamics, swarms, and crowd behavior are described by the FP equation as discussed in Sec. 5):

$$d\boldsymbol{x}(t) = -\nabla\psi(\boldsymbol{x}(t))dt + \sqrt{2\beta^{-1}}dW(t) \tag{5}$$

where $\psi(\cdot) : \mathcal{X} \mapsto \mathbb{R}$ is the potential function, $\beta$ is the inverse temperature in statistical physics, and $W(t)$ is a $d$-dimensional Wiener process. Thus, the change of state involves directed motion down a potential gradient and diffusion resulting in a random walk from the Wiener process. Under finite time step $\Delta t$, the lumped state transition for this SDE follows a Gaussian distribution:

$$T_{\text{MP}}(\boldsymbol{x}'|\boldsymbol{x}) = \left(\frac{\beta}{4\pi\Delta t}\right)^{d/2} \exp\left(\frac{-\beta||\boldsymbol{x}' - \boldsymbol{x} + \nabla\psi(\boldsymbol{x})\Delta t||^2}{4\Delta t}\right). \tag{6}$$

Fully describing this MP transition thus requires $\psi(\cdot)$ and $\beta$. We approach this learning task by enlisting the FP partial differential equation (PDE) that correspondingly describes the evolution of probability density of states $p(\boldsymbol{x})$ under the Itô SDE in Eq. (5):

$$\frac{\partial p(\boldsymbol{x}, t)}{\partial t} = \nabla \cdot (\nabla\psi(\boldsymbol{x})p(\boldsymbol{x}, t)) + \beta^{-1}\Delta p(\boldsymbol{x}, t). \tag{7}$$

As we show below in Sec. 3.6, the form of the FP PDE Eq. (7) can be inferred from data $\mathcal{D}$ using an approach called *Variational System Identification (VSI)*.

### 3.3 Free Energy in an MDP System

After obtaining the MDP transition from Eq. (4), the remaining task for IRL entails estimating the reward function and corresponding optimal policy. This is achieved by a key conjecture of this work, that the value function in MDP is equivalent to the negative potential function in FP of MDP-induced MP by using the free energy functional.

In statistical mechanics, free energy plays a central role in understanding the behavior of physical systems as it allows us to calculate the equilibrium properties and predict the outcomes of (e.g., thermodynamic) processes. The free energy $F$ is defined to be a function of internal energy and entropy of a stochastic system:

$$F(p, \psi) = \int_{\mathcal{X}} \psi(\boldsymbol{x})p(\boldsymbol{x})d\boldsymbol{x} + \beta^{-1}\int_{\mathcal{X}} p(\boldsymbol{x})\log p(\boldsymbol{x})d\boldsymbol{x}. \tag{8}$$

The principle of minimum free energy states that a system will evolve towards a state of minimum $F$ (i.e., maximum stability). Jordan et al. (1997) further proved that the solution of

$$p_{t+1} = \arg\min_p W_2(p_t, p)^2 + \Delta t \, F(p, \psi) \tag{9}$$

converges to the solution of the FP PDE in Eq. (7) as $\Delta t \to 0$, where $W_2(\cdot, \cdot)$ denotes the Wasserstein-2 distance between two distributions. The Wasserstein flows are thus generated by minimizing $F(p, \psi)$ in an isomorphism to the maximization of the value in an MDP.

In an MDP system, the agent's optimal policy is designed to maximize the value function while being constrained by the environment's dynamics (transition function). This means that the agent employs its policy to reach states where the value function is high. By considering the value function as the (negative) potential function, we also observe that the free energy Eq. (8) of an MDP decreases over time in the context of a population of agents or the probabilistic view of the agent's states in the MDP, as shown in Fig. 4 in Appendix B. The MDP system satisfies the principle of minimum energy. Therefore, we propose the following conjecture.

**Conjecture 3.1.** *The state-action value function in a physics-constrained MDP is equivalent to the negative potential function in FP.*

$$Q_\pi(\boldsymbol{s}, \boldsymbol{a}) = -\psi(\boldsymbol{x}); \quad \boldsymbol{x} = [\boldsymbol{s}, \boldsymbol{a}]. \tag{10}$$

*Remark.* The potential function is the driver whose minimization leads to FP dynamics in Eq. (9). The value function is the driver whose maximization leads to the MDP in Eq. (1). The equivalence in Conjecture 3.1 thus leads to the isomorphism between FP dynamics and the MDP. □

The minimum energy can be achieved in two aspects in an MDP. 1) **Learning an optimal policy by** $\arg\min_\psi F(p, \psi)$**:** Any arbitrary policy has its own value function $Q_\pi$ (or potential function $-\psi$) by the *contraction mapping theorem*. Therefore, to minimize the free energy in Eq. (8), the policy should be optimal, and therefore, its corresponding value function should be maximized (by substituting Eq. (2) and (10) into Eq. (8)). 2) **Applying the optimal policy leads to** $\arg\min_p F(p, \psi)$**:** Assuming the agent already adopts an optimal policy in IRL, case 1) above is not considered in our model, but gives us a fundamental reason for why the value function is equivalent to the (negative) FP potential function. If the agent follows the optimal policy, i.e., following the value function (negative potential) gradient, its free energy will decrease over time and finally reach the minimum at the steady-state distribution $p_\infty$ if every state in the MDP is reachable.

### 3.4 THE AGENT'S POLICY CONSTRAINED BY FP

In this section, we show that the Boltzmann policy is the optimal policy for an FP-constrained MDP. The steady-state distribution $p_\infty(\boldsymbol{x})$ of the FP dynamics minimizes the free energy functional, and has the form of the Gibbs-Boltzmann density (Jordan et al., 1997):

$$p_\infty(\boldsymbol{x}) = p_\infty(\boldsymbol{s}, \boldsymbol{a}) = Z^{-1} \exp(-\beta\psi(\boldsymbol{s}, \boldsymbol{a})) = \arg\min_p F(p, \psi) \tag{11}$$

where $Z = \int_\mathcal{S} \int_\mathcal{A} \exp(-\beta\psi(\boldsymbol{s}, \boldsymbol{a})) d\boldsymbol{a} d\boldsymbol{s}$ is a normalization constant. The marginalized steady-state distribution of state $\boldsymbol{s}$ follows:

$$p_\infty(\boldsymbol{s}) = \int_\mathcal{A} p_\infty(\boldsymbol{s}, \boldsymbol{a}) \mathrm{d}\boldsymbol{a} = Z^{-1} \int_\mathcal{A} \exp(-\beta\psi(\boldsymbol{s}, \boldsymbol{a})) d\boldsymbol{a}. \tag{12}$$

Therefore, the steady-state conditional distribution of action $\boldsymbol{a}$ given state $\boldsymbol{s}$ becomes

$$p_\infty(\boldsymbol{a}|\boldsymbol{s}) = \frac{p_\infty(\boldsymbol{s}, \boldsymbol{a})}{p_\infty(\boldsymbol{s})} = \frac{\exp(-\beta\psi(\boldsymbol{s}, \boldsymbol{a}))}{\int_\mathcal{A} \exp(-\beta\psi(\boldsymbol{s}, \boldsymbol{a}')) d\boldsymbol{a}'}, \tag{13}$$

which has the same form as the Boltzmann policy

$$\pi(\boldsymbol{a}|\boldsymbol{s}) = \frac{\exp(\beta Q_\pi(\boldsymbol{s}, \boldsymbol{a}))}{\int_\mathcal{A} \exp(\beta Q_\pi(\boldsymbol{s}, \boldsymbol{a}')) d\boldsymbol{a}'} \tag{14}$$

in previous RL and IRL studies (Sallans and Hinton, 2004; Ziebart et al., 2010; Haarnoja et al., 2018a;b; Skalse and Abate, 2023), thus providing some evidence for our Conjecture 3.1.

In Sec. 3.2, we have shown that an MDP reduces to an MP when the policy is fixed. Now, we expand the MP back to an MDP. For the lumped state $\boldsymbol{x}$, the transformation of $p_t(\boldsymbol{s})$ to $p_t(\boldsymbol{s}, \boldsymbol{a})$ happens through the optimal policy, which follows the Boltzmann distribution. While the optimal policy has been identified as Boltzmann in the steady state, a reasonable assumption is that this result holds also in the transient state, as it is consistent with the conclusion of the time-invariant policy in the infinite-horizon MDP.

We discuss how the Boltzmann policy is optimal in FP-constrained MDP in the transient state as well under certain mild conditions. The Wasserstein distance, $W_2(\cdot)$ in Eq. (9) appears in a form known as a movement-limiter. It imposes a physics constraint that the change in the distribution should be small and approaches zero over an infinitesimal time step. In this limit, it thus can be neglected in the context of the MDP, and the minimization in Eq. (9) becomes $\arg\min_p F(p, -Q_\pi)$ by substituting Eq. (10) into Eq. (9). Note that $p(\boldsymbol{x}) = p(\boldsymbol{s})\pi(\boldsymbol{a}|\boldsymbol{s})$, and because $p(\boldsymbol{s})$ is obtained from the previous time step via the transition function (environment) and therefore cannot be optimized, the optimization problem becomes that finding an optimal policy with minimum free energy:

$$\arg\min_{\pi \in \Pi} \int_\mathcal{S} p(\boldsymbol{s}) \int_\mathcal{A} \pi(\boldsymbol{a}|\boldsymbol{s}) \left[ -Q_\pi(\boldsymbol{s}, \boldsymbol{a}) + \beta^{-1} \log \pi(\boldsymbol{a}|\boldsymbol{s}) \right] d\boldsymbol{a} d\boldsymbol{s} = Z_a^{-1} \exp(\beta Q_\pi(\boldsymbol{s}, \boldsymbol{a})) \tag{15}$$

where $Z_a = \int_\mathcal{A} \exp(\beta Q_\pi(\boldsymbol{s}, \boldsymbol{a}')) d\boldsymbol{a}'$. The detailed derivation is provided in Appendix C.

### 3.5 INVERSE BELLMAN EQUATION

With the transition function $T(\cdot)$ of the MDP by Eq. (4), state-action value function $Q_\pi(\boldsymbol{s}, \boldsymbol{a})$ by Eq. (10) and policy $\pi(\cdot)$ by Eq. (15) obtained from FP equation discussed in Sec. 3.2 to 3.4, the reward function $R(\cdot)$ can be simply derived from the inverse Bellman equation:

$$R(\boldsymbol{s}, \boldsymbol{a}) = Q_\pi(\boldsymbol{s}, \boldsymbol{a}) - \gamma \mathbb{E}_{\boldsymbol{s}' \sim T(\cdot|\boldsymbol{s}, \boldsymbol{a}), \boldsymbol{a}' \sim \pi(\cdot|s')} \left[ Q_\pi(\boldsymbol{s}', \boldsymbol{a}') \right]. \tag{16}$$

Hence, there is a unique reward function Eq. (16) corresponding to a pair of transition and value functions as shown in Theorem 3.2.

**Theorem 3.2.** *Define the inverse Bellman operator* $\mathcal{T} : \mathcal{Q} \mapsto \mathcal{R}$ *(where* $\mathcal{Q}, \mathcal{R}$ *denote the spaces of value functions and reward functions, respectively) such that*

$$(\mathcal{T} \circ Q_\pi)(\boldsymbol{s}, \boldsymbol{a}) = Q_\pi(\boldsymbol{s}, \boldsymbol{a}) - \gamma \mathbb{E}_{\boldsymbol{s}' \sim T(\cdot|\boldsymbol{s},\boldsymbol{a}), \boldsymbol{a}' \sim \pi(\cdot|s')} \left[ Q_\pi(\boldsymbol{s}', \boldsymbol{a}') \right]. \tag{17}$$

*For a transition* $T(\cdot)$ *Eq. (4) and policy* $\pi(\cdot)$ *Eq. (14),* $\mathcal{T}$ *is a bijective mapping.*

*Sketch of proof.* We prove that the discretized Bellman operator: $\mathcal{T} \circ Q_\pi = (\mathbb{I} - \gamma T)Q_\pi$ is linear operator with a invertible matrix. See Appendix D or Garg et al. (2021) for the complete proof. □

This leads to the conclusion that estimating the potential function $\psi(\cdot)$ in the FP equation corresponding to the induced MP is sufficient to infer the reward function in the MDP.

### 3.6 INFERENCE OF THE FOKKER-PLANCK PDE

We use VSI method for data-driven inference of the FP PDE. Readers are directed to Appendix E and Wang et al. (2019; 2021) for background and details on VSI. We consider the spatiotemporal state-action density field, $p(\boldsymbol{x}, t)$ with $(\boldsymbol{x}, t) \in \Omega \times [0, \tau]$ where $\Omega$ is the continuous domain of admissible state-action values and $[0, \tau]$ is the time interval. The weak form of FP PDE Eq. (7) with periodic boundary conditions:

$$\int_{\mathcal{S} \times \mathcal{A}} \frac{\partial p}{\partial t} w d\Omega + \int_{\mathcal{S} \times \mathcal{A}} p \nabla \psi \cdot \nabla w + \beta^{-1} \nabla p \cdot \nabla w d\Omega = 0 \tag{18}$$

where $w$ is the weighting function commonly used in variational calculus. Noting that the $\boldsymbol{x} = (s_1, \cdots, s_{d_s}, a_1, \cdots, a_{d_a})$, we consider a tensor basis for interpolating the *unknown potential function*, $\psi$:

$$\psi(\boldsymbol{x}) = \sum_{i_1, \cdots, i_d} \theta_{i_1, \cdots, i_d} \phi_{i_1, \cdots, i_d}(\boldsymbol{x}), \qquad \phi_{i_1, \cdots, i_d}(\boldsymbol{x}) = \Pi_{k=1 \cdots d} h_{i_k}(x_k) \tag{19}$$

where $h_i$ represents 1-d *Hermite cubic* functions with added periodicity. The weak form leads to the following residual:

$$\mathcal{R} = \int_{\mathcal{S} \times \mathcal{A}} \frac{\partial p}{\partial t} w d\Omega + \sum_{i_1, \cdots, i_d} \theta_{i_1, \cdots, i_d} \int_{\mathcal{S} \times \mathcal{A}} p \nabla \phi_{i_1, \cdots, i_d} \cdot \nabla w d\Omega + \beta^{-1} \int_{\mathcal{S} \times \mathcal{A}} \nabla p \cdot \nabla w d\Omega. \tag{20}$$

The parameters, $\boldsymbol{\theta} \equiv \{\theta_{i_1, \cdots, i_d}\}_{i_1, \cdots, i_d}$ are estimated using the data field, $p^{\text{data}}(\boldsymbol{x}, t)$ evaluated at discrete timesteps $t \in \{t_1, \cdots, t_n\}$

$$\boldsymbol{\theta}^* = \arg\min_{\boldsymbol{\theta}} \sum_{t \in \{t_1, \cdots, t_n\}} ||\mathcal{R}(\boldsymbol{p}^{\text{data}}(., \boldsymbol{t}); \boldsymbol{\theta})||_2^2. \tag{21}$$

In favor of a parsimonious model, which can be quantified as the sparsity of basis terms, we intend to estimate the most significant terms in the prescribed ansatz for $\psi$ and drop all the insignificant ones. A popular greedy approach is the *stepwise regression method*. In this approach, we iteratively identify a term that, when eliminated, causes a minimal change in the loss of the reduced optimization problem. To avoid dropping more than the necessary terms, we perform the statistical *F-test* that signifies the relative change in loss with respect to the change in the number of terms. Therefore, we use a threshold for the F-value as a stopping criterion for stepwise regression. More details on this approach are available in the previous works mentioned above.

## 4 EXPERIMENTS

In this section, we demonstrate our method on a synthetic example and a biological problem of cancer cell metastasis. FP-IRL is not directly applicable to off-the-shelf RL benchmarks (e.g., OpenAI Gym problems) because their state-action pairs do not necessarily follow the FP dynamic. However, we provide the *Mountain Car* problem with a modified dynamic in Appendix F.2 as an additional example. All experiments were conducted on the Expanse cluster resource provided by NSF's Access program. Each training experiment utilized single CPU nodes (AMD EPYC 7742). The memory required for each experiment depends on the discretizations $n$ and scales with $\mathcal{O}(n^d)$.

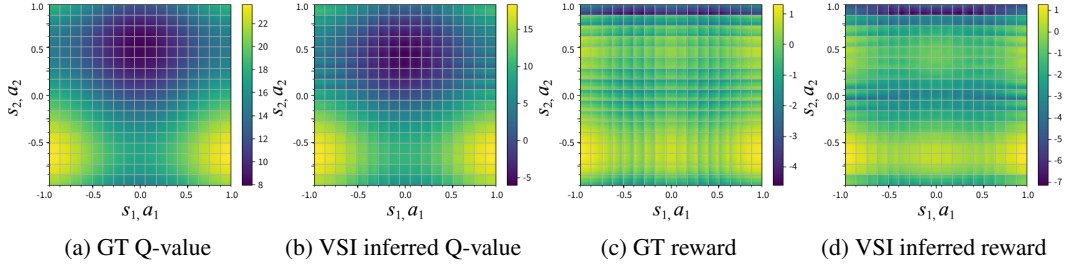

(a) GT Q-value  (b) VSI inferred Q-value  (c) GT reward  (d) VSI inferred reward

Figure 1: Comparison of inferred ground truth value and reward (using highest resolution mesh with $N = 17$) with respect to its ground truth. We show that the bias in value estimation (e.g., between (a) and (b)) does not affect the transition and policy inference and consequently the reward estimation in Appendix F.3. Value and reward functions corresponding to the 4d state-action variable are displayed using larger grids for state variables and sub-grids for action variables. The color represents the function value of state-action (e.g., $Q(\boldsymbol{s}, \boldsymbol{a})$, $R(\boldsymbol{s}, \boldsymbol{a})$).

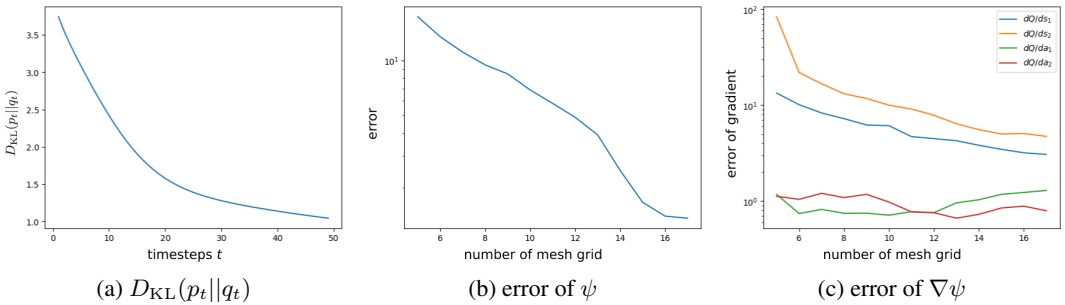

(a) $D_{\mathrm{KL}}(p_t\|q_t)$  (b) error of $\psi$  (c) error of $\nabla\psi$

Figure 2: (a) KL divergence $D_{\mathrm{KL}}(p_t\|q_t)$ of the probability distribution between data distribution and simulated probability distribution using inferred policy and transition. The errors of the (b) value function and (c) gradient of the value function, estimated as $\left(\frac{1}{|\Omega|}\int_\Omega (f(\boldsymbol{x}) - f_{\mathrm{GT}}(\boldsymbol{x}))^2 d\boldsymbol{x}\right)^{1/2}$.

## 4.1 SYNTHETIC EXAMPLE AND CONVERGENCE STUDY

The purpose of the synthetic example is to provide validation against known ground truth, and to carry out a convergence study. We first define a value function $Q(\cdot)$, as shown in Fig. 1a, using the Hermite orthogonal polynomial basis to have sufficient expressivity. The transition, optimal policy, and reward (in Fig. 1c) are then induced from Eq. (4), Eq. (14), Eq. (16), respectively. Starting with a initial distribution of $p_0(\boldsymbol{s}) \propto 1/(\sin^2(4\pi s_1) + \sin^2(4\pi s_2) + 1)$, we estimated the probability distribution over all timesteps, $\mathcal{D}_p = \{p_t^{\mathrm{data}}(\boldsymbol{s}, \boldsymbol{a})\}_t$, via evolution by discretized MDP transition. Alternately, one can sample trajectories $\mathcal{D} = \left\{(\boldsymbol{s}_t^{(i)}, \boldsymbol{a}_t^{(i)})_{t=0}^\tau\right\}_{i=1}^m$ and estimate the probability densities from them. Finally, we estimated the value function using VSI. We show that our method can accurately estimate the value function and reward function in Fig. 1b and 1d, respectively, when using a high-resolution mesh for state-action space. In Fig. 2a, we show that the *Kullback–Leibler (KL) divergence* $D_{\mathrm{KL}}(p_t\|q_t)$ between the probability distribution $p_t$ in data $\mathcal{D}_p$, and the probability distribution $q_t$ simulated by using inferred optimal policy and transition, is decreasing with time, alluding to convergence to the same steady state. However, predicting the transient behavior is much more challenging.

We also consider the effect of the mesh resolution of the space $\mathcal{S} \times \mathcal{A}$. Previous studies (Wang et al., 2019; 2021) have shown convergence in the inference conducted using VSI method. Here we investigate the convergence in the state-action value function and, consequently, the reward. We consider a box domain with $\Omega = [-1, 1]^4$ using Cartesian meshes with nodes at $\boldsymbol{x} \in \left\{-1, -1 + \frac{2}{N}, \cdots, -1 + \frac{2i}{N}, \cdots, 1\right\}^4$. We evaluate the error estimated $\psi$ compared to the *ground truth* state-action value generated using fixed-point iteration (details provided in Appendix E). The results of the convergence analysis of value function $\psi$ and its gradient $\nabla\psi$ are presented in Fig. 2b and 2c where the error is observed to decrease with finer mesh resolution.

### 4.2 CANCER CELL METASTASIS

As a proof-of-concept with real-world data, we apply our algorithm to an experiment dataset (including 1332 cells in 361 timesteps) of MDA-MB231 cancer cells in a migration assay Fig. 3a (Ho et al., 2022), whose dynamics is widely understood to be governed in the continuous limit by different versions of FP equations. A chemical gradient of the chemo-attractant CXCL12 is applied pointing to the left: the negative horizontal direction. This induces the cells to migrate leftward, on average. The cancer cell is modeled as a decision-making agent under the mathematical formalism of an MDP. The observed data reflects the agent choosing the optimal state-dependent action to maximize its expected cumulative reward while navigating under the constraints of its environment. Given this foundation, we aim to identify 1) the reward and 2) the policy and transition from the trajectories. The reward represents our hypothesis, motivated by the emerging understanding of the cancer biology community, that the cells' diversity of response could be understood in terms of them optimizing a function that is as yet unknown. Learning the transition and policy can help predict cell behavior. We define the velocity $[v_x, v_y]^\top$ as state variables, and [Akt, ERT]$^\top$ signaling as the action variables. The data is rescaled to $[-1, 1]^d$, and we empirically estimate the probability density of cells. Our FP-IRL algorithm applied to this dataset recovers the result that the cell will receive a high reward for moving leftward with a high velocity in agreement with our knowledge about the experimental setup, as shown in Fig. 3b. Interestingly, FP-IRL also uncovers a vertical component to the velocity providing high rewards. FP-IRL infers a policy expressing low Akt when moving towards left with high speed to be optimal as shown in Fig. 3c. More discussion on these results is provided in Appendix F.4.

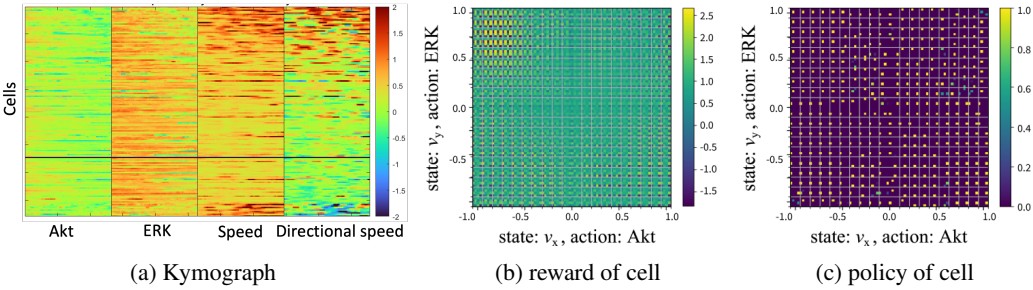

| (a) Kymograph | (b) reward of cell | (c) policy of cell |
| --- | --- | --- |

Figure 3: (a) Kymograph for the cancer cell migration data: each column shows the different variables measured from that experiment, and each row shows the measurement value over time. (b) reward and (c) policy inferred from data.

## 5 DISCUSSION

**Significance**   In this work, we have conjectured an isomorphism between the FP-governed physics and MDPs. On this basis, we have proposed a novel physics-constrained IRL algorithm and demonstrated it on a problem studying the dynamics of living agents in biology. As one example, this approach could initiate a new paradigm of scientific machine learning for physics-based cancer biology, in which revealing the deemed reward gained by cell agents can allow us to rationalize their behavior. In particular, the injection of physics principles allows IRL to proceed without relying on empirical estimation of transition functions. Combining physics and IRL in such manner is novel and has not been explored previously. **Interpretability in physics:** Processes whose time-continuous form governed by the FP equation are of fundamental interest in this work. Therefore, by first inferring the governing FP equation (via VSI) and using Conjecture 3.1, we obtain a value function that has an unambiguous interpretation in terms of physics. From this form of result, we can extract terms reflecting physics mechanisms such as drift, diffusion, and sources/sinks. **Combating ill-posedness:** IRL is inherently ill-posed since there exist many combinations of reward and transition that can fit the demonstrated trajectories. The empirically estimated transition in conventional IRL approaches may not inherit the underlying dynamics, and is often challenging to generalize to state and action regions away from training samples relying on data alone. By constraining with FP dynamics, we systematically reduce this ambiguity to identify a unique pair of transition and reward functions that comply with the FP dynamics. **Computation efficiency:** When

physics constraints are not imposed, the searches for reward and transition have to cover a larger space and therefore more expensive. Additionally, existing IRL approaches typically involve an outer loop of reward search coupled with an inner loop of policy optimization (forward RL), which is very compute-intensive. FP-IRL avoids such iterations altogether and instead induces a regression problem leveraging the FP physics that is also computationally more stable. **Applicability:** With physics-constrained modeling, this allows the application of FP-IRL to problems where the transition is not available and has not been mathematically modeled, or discovered. Cancer cell migration, as well as the migration of other cell types, is known to be governed by physics, specifically that described by the FP equation (Bressloff, 2014). Therefore, there is interest in the fields of biology, biophysics, and physics more broadly, to have scientific machine learning methods that respect these physics. We achieve this by combining machine learning ideas (IRL) with physics principles (Minimum Energy Principle and FP dynamics). Although the proposed method may not apply directly to some RL problem domains, such as robotics, many other physics phenomena encompassing Brownian dynamics (Keilson and Storer, 1952), swarming (Correll and Hamann, 2015) and crowd behavior (Dogbé, 2010), pattern formation and morphogenesis (Garikipati, 2017) are also described by FP equations in the continuous limit, and this work would also be applicable to them.

**Limitations**   One limitation of FP-IRL is that it is formed based on the free energy in FP dynamics; the target dynamics therefore must submit to this description in the continuous limit. The SDE constrains the state and action space $\mathcal{S} \times \mathcal{A}$ to be $\mathbb{R}^n$, where we have assumed periodic boundary conditions on the dynamics. Also, we use PDEs, limiting the definition of state and action variable to be continuous. The convergence analysis shows that a finer discretization is required to accurately estimate the potential function and therefore reward function. This makes our method less suitable for coarsely binned state-action spaces. This method can extend to high-dimensional state-action spaces. However, having its root in finite element methods (FEM), it similarly suffers from the curse of dimensionality. Alternatively, the FEM basis formulation can be extended to neural network-based methods for approximating the value function. To recognize whether a chosen system follows FP dynamics requires some prior domain knowledge. Finally, our method rests on mean-field physics, and therefore, may not be suitable to study multi-agent systems with interactions in the current setting.

**Future Work**   There are several directions in which this physics-based framework for IRL can be extended. Possible theoretical extensions include: (1) more expressive diffusive mechanisms like Maxwell-Stefan diffusion that account for the interaction of agents (e.g., collisions between agents) and (2) considerations for reflected Brownian motion that describes the evolution of agents in bounded domains. Turning toward additional capabilities for this framework, many physical systems, such as migration mechanics of cells, naturally involve the proliferation and death of these individual agents. Modeling such mechanisms that involve terminal and source states in MDP results in reactive mechanisms in FP dynamics. This extension will also be a subject of consequent studies. Furthermore, we observe a correlation between the *Markov Potential Game* and the concept of potential in free energy functional. Another possible future work is an extension to multi-agent problems and uncovering the inter-agent rewards.

## 6   CONCLUSION

We developed a novel physics-based IRL algorithm, FP-IRL, that can uncover both the reward function and transition function even when confronted with limited information about the system under investigation. Our approach leverages the fundamental physics principle of minimum energy and establishes a conjecture regarding the structural isomorphism between FP and MDP. With the conjecture, we can estimate the reward and transition with low computational expense. We validate the efficacy of our method in a synthetic problem and show that it converges to the true solution as we enhance the resolution of the mesh. Finally, we employ our algorithm to infer the reward structure for dynamics of kinase-dependent migration of cancer cells from real-life experiment data.

**Reproducibility Statement**   Our methods for inference (VSI) and synthetic experiment data generation are detailed in sections Appendices E and F.1, respectively, facilitating reproducibility. For further reproducibility, our code will be made available via an anonymous repository link.

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

# SUPPLEMENTAL MATERIALS

## A  SUMMARY OF FP-IRL ALGORITHM

The FP-IRL method is summarized in Algorithm 1. We first transform the MDP into an MP, allowing us to connect it to the FP PDE. We then use VSI to estimate the potential function and transition probability function of the system. Leveraging our conjecture, the reward and policy in MDP can be subsequently estimated from the learned potential function with minimal computational cost.

---

**Algorithm 1:** FP-IRL

---

**Input:** Markov decision process without reward and transition functions $\mathcal{M}/\{R, T\}$, observed trajectories $\mathcal{D}$.
**Output:** Estimated reward $R$, policy $\pi$, and transition function.
Use VSI to estimate the potential function $\psi(\boldsymbol{x})$ by solving Eq. (21) ;
Estimate transition $T(\boldsymbol{s}'|\boldsymbol{s}, \boldsymbol{a})$ using Eq. (3) ;
Estimate policy $\pi(\boldsymbol{a}|\boldsymbol{s})$ by Boltzmann policy Eq. (14) ;
Estimate reward $R(\boldsymbol{s}, \boldsymbol{a})$ by Eq. (16).

---

## B  ILLUSTRATION OF MINIMIZATION OF FREE ENERGY IN AN MDP

We simulate the probability density evaluation of an MDP system, and evaluate its free energy over time by substituting Eq. (10) into Eq. (8). In Fig. 4, we show that the free energy of an MDP system decreases over time, eventually reaching its minimum at the steady-state distribution. This depicts that the MDP also follows the energy minimization principle, thus providing evidence to our Conjecture 3.1: the negative value function is equated to the potential function.

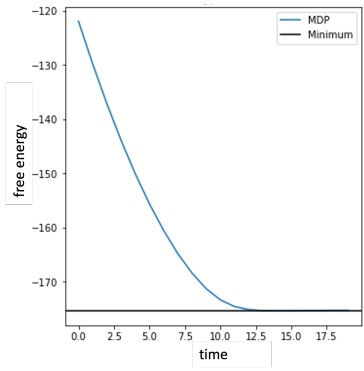

Figure 4: Free energy of an MDP system decreases over time.

## C  THE AGENT'S POLICY CONSTRAINED BY FP

In this section, we show that the Boltzmann policy is the optimal policy in the FP-contrained MDP.

First recall the chain rule of entropy (Cover and Thomas, 2012):

$$H(\mathbf{s}, \mathbf{a}) = H(\mathbf{s}) + H(\mathbf{a}|\mathbf{s}). \tag{22}$$

*Proof.*

$$H(\mathbf{s}, \mathbf{a}) = - \int_{\mathcal{S}} \int_{\mathcal{A}} p(\boldsymbol{s}, \boldsymbol{a}) \log p(\boldsymbol{s}, \boldsymbol{a}) d\boldsymbol{a} \, d\boldsymbol{s} \tag{23}$$

$$= - \int_{\mathcal{S}} \int_{\mathcal{A}} p(\boldsymbol{s})\pi(\boldsymbol{a}|\boldsymbol{s}) \log\left(p(\boldsymbol{s})\pi(\boldsymbol{a}|\boldsymbol{s})\right) d\boldsymbol{a}\, d\boldsymbol{s} \tag{24}$$

$$= - \int_{\mathcal{S}} \int_{\mathcal{A}} p(\boldsymbol{s})\pi(\boldsymbol{a}|\boldsymbol{s})[\log p(\boldsymbol{s}) + \log \pi(\boldsymbol{a}|\boldsymbol{s})]d\boldsymbol{a}\, d\boldsymbol{s} \tag{25}$$

$$= - \int_{\mathcal{S}} \int_{\mathcal{A}} p(\boldsymbol{s})\pi(\boldsymbol{a}|\boldsymbol{s}) \log p(\boldsymbol{s})d\boldsymbol{a}\, d\boldsymbol{s} - \int_{\mathcal{S}} \int_{\mathcal{A}} p(\boldsymbol{s})\pi(\boldsymbol{a}|\boldsymbol{s}) \log \pi(\boldsymbol{a}|\boldsymbol{s})d\boldsymbol{a}\, d\boldsymbol{s} \tag{26}$$

$$= - \int_{\mathcal{S}} p(\boldsymbol{s}) \log p(\boldsymbol{s}) \int_{\mathcal{A}} \pi(\boldsymbol{a}|\boldsymbol{s})d\boldsymbol{a}\, d\boldsymbol{s} - \int_{\mathcal{S}} p(\boldsymbol{s}) \int_{\mathcal{A}} \pi(\boldsymbol{a}|\boldsymbol{s}) \log \pi(\boldsymbol{a}|\boldsymbol{s})d\boldsymbol{a}\, d\boldsymbol{s} \tag{27}$$

$$= - \int_{\mathcal{S}} p(\boldsymbol{s}) \log p(\boldsymbol{s})d\boldsymbol{s} - \int_{\mathcal{S}} p(\boldsymbol{s}) \int_{\mathcal{A}} \pi(\boldsymbol{a}|\boldsymbol{s}) \log \pi(\boldsymbol{a}|\boldsymbol{s})d\boldsymbol{a}\, d\boldsymbol{s} \tag{28}$$

$$= H(\mathbf{s}) + H(\mathbf{a}|\mathbf{s}). \tag{22}$$

$\square$

We then substitute Eq. (22) into Eq. (8):

$$F(p, -Q_\pi) = - \int_{\mathcal{S}} \int_{\mathcal{A}} p(\boldsymbol{s})p(\boldsymbol{a}|\boldsymbol{s})Q_\pi(\boldsymbol{s}, \boldsymbol{a})d\boldsymbol{a}\, d\boldsymbol{s} - \beta^{-1}[H(\mathbf{s}) + H(\mathbf{a}|\mathbf{s})] \tag{29}$$

$$= \int_{\mathcal{S}} p(\boldsymbol{s}) \int_{\mathcal{A}} \pi(\boldsymbol{a}|\boldsymbol{s})[-Q_\pi(\boldsymbol{s}, \boldsymbol{a}) + \beta^{-1} \log \pi(\boldsymbol{a}|\boldsymbol{s})]d\boldsymbol{a}d\boldsymbol{s} + \beta^{-1} \int_{\mathcal{S}} p(\boldsymbol{s}) \log p(\boldsymbol{s})d\boldsymbol{s}. \tag{30}$$

Because $p(\boldsymbol{s})$ is obtained from the previous time step via the transition function (i.e., environment) and therefore cannot be optimized, the optimization problem of $\arg\min_p F(p, -Q)$ becomes one of finding the optimal policy that minimizes the free energy:

$$\pi^*(\cdot) = \arg\min_{\pi \in \Pi} \int_{\mathcal{S}} p(\boldsymbol{s}) \int_{\mathcal{A}} \pi(\boldsymbol{a}|\boldsymbol{s}) \left[-Q_\pi(\boldsymbol{s}, \boldsymbol{a}) + \beta^{-1} \log \pi(\boldsymbol{a}|\boldsymbol{s})\right] d\boldsymbol{a}d\boldsymbol{s} = Z_a^{-1} \exp(\beta Q_\pi(\boldsymbol{s}, \boldsymbol{a})) \tag{15}$$

where $Z_a = \int_{\mathcal{A}} \exp(\beta Q_\pi(\boldsymbol{s}, \boldsymbol{a}'))d\boldsymbol{a}'$.

## D  INVERSE BELLMAN OPERATOR

In this section, we provide the proof for Theorem 3.2.

Note that the proof is similar to the proof of Lemma 3.1 in Appendix 2 of (Garg et al., 2021), but their inverse Bellman operator is defined as

$$R(\boldsymbol{s}, \boldsymbol{a}) = (\mathcal{T}Q_\pi)(\boldsymbol{s}, \boldsymbol{a}) = Q_\pi(\boldsymbol{s}, \boldsymbol{a}) - \gamma \mathbb{E}_{\substack{\boldsymbol{s}' \sim T(\cdot|\boldsymbol{s}, \boldsymbol{a}), \\ \boldsymbol{a}' \sim \pi(\cdot|\boldsymbol{s}')}} [Q_{\pi,\mathrm{soft}}(\boldsymbol{s}', \boldsymbol{a}') - \log \pi(\boldsymbol{a}'|\boldsymbol{s}')] \tag{31}$$

where $Q_{\pi,\mathrm{soft}}(\cdot)$ is a so-called soft Bellman equation, while ours is defined as

$$R(\boldsymbol{s}, \boldsymbol{a}) = (\mathcal{T}Q_\pi)(\boldsymbol{s}, \boldsymbol{a}) = Q_\pi(\boldsymbol{s}, \boldsymbol{a}) - \gamma \mathbb{E}_{\substack{\boldsymbol{s}' \sim T(\cdot|\boldsymbol{s}, \boldsymbol{a}), \\ \boldsymbol{a}' \sim \pi(\cdot|\boldsymbol{s}')}} [Q_\pi(\boldsymbol{s}', \boldsymbol{a}')] \tag{32}$$

where $Q_\pi(\boldsymbol{s}, \boldsymbol{a})$ denotes the conventional Bellman expectation function of a policy $\pi(\cdot)$.

**Lemma D.1.** *The matrix $(I - A)$ is nonsingular if the norm of matrix $A$ is less than 1 (i.e. $||A|| < 1$).*

*Proof.* Proof by contradiction: Let $I - A$ be singular. Therefore, there exists an $\boldsymbol{x}$ (where $\boldsymbol{x} \neq 0$) such that $(I - A)\boldsymbol{x} = 0$. Then, $||\boldsymbol{x}|| = ||A\boldsymbol{x}|| \leq ||A||\, ||\boldsymbol{x}||$, and therefore $||A|| \geq 1$, which contradicts the ansatz $||A|| < 1$. Therefore $(I - A)$ is nonsingular and invertible. $\square$

**Theorem 3.2.** *Define the inverse Bellman operator $\mathcal{T} : \mathcal{Q} \mapsto \mathcal{R}$ (where $\mathcal{Q}, \mathcal{R}$ denote the spaces of value functions and reward functions, respectively) such that*

$$(\mathcal{T} \circ Q_\pi)(\boldsymbol{s}, \boldsymbol{a}) = Q_\pi(\boldsymbol{s}, \boldsymbol{a}) - \gamma \mathbb{E}_{\boldsymbol{s}' \sim T(\cdot|\boldsymbol{s}, \boldsymbol{a}), \mathbf{a}' \sim \pi(\cdot|s')} [Q_\pi(\boldsymbol{s}', \boldsymbol{a}')]. \tag{17}$$

*For a transition $T(\cdot)$ Eq. (4) and policy $\pi(\cdot)$ Eq. (14), $\mathcal{T}$ is a bijective mapping.*

*Proof.* For a fixed transition probability function $T(\boldsymbol{s}'|\boldsymbol{s},\boldsymbol{a})$, and a fixed policy probability function $\pi(\boldsymbol{a}|\boldsymbol{s})$ in MDP, the joint transition probability function $T_\pi(\boldsymbol{s}',\boldsymbol{a}'|\boldsymbol{s},\boldsymbol{a}) = T(\boldsymbol{s}'|\boldsymbol{s},\boldsymbol{a})\pi(\boldsymbol{a}'|\boldsymbol{s}')$ is fixed as well. The inverse Bellman operator can be denoted in matrix form in the discrete case:

$$\boldsymbol{r} = \boldsymbol{q} - \gamma\boldsymbol{T}_\pi\boldsymbol{q} = (\boldsymbol{I} - \gamma\boldsymbol{T}_\pi)\boldsymbol{q} \tag{33}$$

where $\boldsymbol{r} \in \mathbb{R}^{n_s \cdot n_a}$ denotes reward vector, $\boldsymbol{q} \in \mathbb{R}^{n_s \cdot n_a}$ denotes state-action value vector, $\boldsymbol{T}_\pi \in \mathbb{R}^{(n_s \cdot n_a) \times (n_s \cdot n_a)}$ denotes the joint transition matrix, and $n_s = |\mathcal{S}|, n_a = |\mathcal{A}|$ denotes the number of discretized states and actions, respectively. $(\boldsymbol{I} - \gamma\boldsymbol{T}_\pi)$ is invertible because $||\gamma\boldsymbol{T}_\pi||_1 < 1$ (because $T_\pi$ denotes a probability function, i.e. $||\boldsymbol{T}_\pi||_1 = 1$, and $\gamma \in [0,1)$), as shown in Lemma D.1 . Therefore, because the inverse Bellman operator is a linear transformation with an invertible square transformation matrix $I - \gamma T$, the inverse Bellman operator $\mathcal{T}$ is a bijection when $T(\cdot), \pi(\cdot)$ are fixed. $\qquad\square$

# E VARIATIONAL SYSTEM IDENTIFICATION

In this section, we present the details for the VSI method in Sec. 3.6

## E.1 FINITE ELEMENT INTERPOLATION

We consider a d-dimensional hypercube domain, $\Omega = \Pi_{i=\{1,\cdots,d\}}[a_i, b_i] \subset \mathbb{R}^d$. A partition of $\Omega$ into elements $\Omega_e$ is constructed by first partitioning the line segment $[a_i, b_i]$ along each dimension as $[a_i, b_i] = \cup_{j=1}^{k_i}[x_i^j, x_i^{j+1}]$ with $x_i^1 = a_i$, $x_i^{k_i} = b_i$ and $x_i^j < x_i^{j+1}$. Finally, the d-dimensional hypercube element is constructed by taking the tensor product of the grid points as $\Omega_{e=(i_1,\cdots,i_d)} = \Pi_l[x_l^{i_l}, x_l^{i_l+1}]$. In the finite element formulation presented here, all function values are known at the grid points and the values within the element are interpolated from the neighbouring grid points as $p(\boldsymbol{x}) = \sum_{r=1}^{2^d} p_{e(r)}N_r(\boldsymbol{x})$. Here $p$ represents the function being interpolated with $\boldsymbol{x}$ inside element, $e$, and $p_{e(r)}$ being the value at the $r^{th}$ neighbour of the $e^{th}$ element. The shape functions, $N_r$ are constructed using the tensor product of linear Lagrange interpolations in each dimension. We first linearly map coordinates of each element onto a unit hypercube i.e. $\Omega_e \rightarrow [0,1]^d$, representing the new coordinates with $\boldsymbol{\xi}$. As an example, the Lagrange tensor product basis functions for 4-d case in this case are given as follows:

$$N_1 = (1 - \xi_1)(1 - \xi_2)(1 - \xi_3)(1 - \xi_4)$$
$$N_2 = (\xi_1)(1 - \xi_2)(1 - \xi_3)(1 - \xi_4)$$
$$N_3 = (1 - \xi_1)(\xi_2)(1 - \xi_3)(1 - \xi_4)$$
$$N_4 = (\xi_1)(\xi_2)(1 - \xi_3)(1 - \xi_4)$$
$$N_5 = (1 - \xi_1)(1 - \xi_2)(\xi_3)(1 - \xi_4)$$
$$N_6 = (\xi_1)(1 - \xi_2)(\xi_3)(1 - \xi_4)$$
$$N_7 = (1 - \xi_1)(\xi_2)(\xi_3)(1 - \xi_4)$$
$$N_8 = (\xi_1)(\xi_2)(\xi_3)(1 - \xi_4)$$
$$N_9 = (1 - \xi_1)(1 - \xi_2)(1 - \xi_3)(\xi_4)$$
$$N_{10} = (\xi_1)(1 - \xi_2)(1 - \xi_3)(\xi_4)$$
$$N_{11} = (1 - \xi_1)(\xi_2)(1 - \xi_3)(\xi_4)$$
$$N_{12} = (\xi_1)(\xi_2)(1 - \xi_3)(\xi_4)$$
$$N_{13} = (1 - \xi_1)(1 - \xi_2)(\xi_3)(\xi_4)$$
$$N_{14} = (\xi_1)(1 - \xi_2)(\xi_3)(\xi_4)$$
$$N_{15} = (1 - \xi_1)(\xi_2)(\xi_3)(\xi_4)$$
$$N_{16} = (\xi_1)(\xi_2)(\xi_3)(\xi_4).$$

## E.2 RESIDUE EVALUATION

The finite element interpolation results in following form of the residual, which is linear in the PDE parameters, $\beta^{-1}, \theta_{(j_1,\ldots,j_d)}$:

$$\mathcal{R} = \boldsymbol{y} - [\boldsymbol{\Xi}_0, \cdots, \boldsymbol{\Xi}_{(j_1,\cdots,j_d)}, \cdots] \cdot [\beta^{-1}, \cdots, \theta_{(j_1,\cdots,j_d)}, \cdots]$$

where each entry of the vectors $\boldsymbol{y}$ and $\Xi$ is evaluated for each timestep. The components of $\boldsymbol{y}$, $\Xi$ and $\Xi_0$ are:

$$y = \sum_e \sum_{r=1}^{2^d} \int_{\Omega_e} \frac{\partial p_{e(r)}}{\partial t} N_{e(r)} w d\Omega$$

$$\Xi_0 = \sum_e \sum_{r=1}^{2^d} \int_{\Omega_e} p_{e(r)} \nabla_x N_{e(r)} \cdot \nabla_x w d\Omega$$

$$\Xi_{(j_1, \cdots, j_d)} = \sum_e \sum_{r=1}^{2^d} \int_{\Omega_e} p_{e(r)} N_{e(r)} \nabla_x \phi_{(j_1, \cdots, j_d)} \cdot \nabla_x w d\Omega$$

where $w \in \{\overline{N_1}, \cdots, \overline{N}_{k1 \times \cdots \times k_d}\}$ with each each $\overline{N}_i$ representing the finite element interpolation of a function that is 1 on $i^{th}$ node and 0 on every other node. The integrations are efficiently evaluated using Gauss-Legendre integration method.

### E.3    HERMITE CUBIC INTERPOLATIONS FOR $\psi$

We construct a parameterization for a $d$-dimensional differentiable function as:

$$\phi_{j_1, \cdots, j_d}(\boldsymbol{x}) = h_{j_1}(x_1) \times \cdots \times h_{j_d}(x_d)$$

where $h_k$ represents the Hermite cubic interpolation along each dimension. This interpolation scheme is based on piecewise cubic polynomials and provides a smooth representation of $\phi_{j_1, \cdots, j_d}(\boldsymbol{x})$. In a 1d Hermite cubic interpolation of a function, for instance, $f(x) = \sum_k \theta_k h_k(x)$, the parameters $\theta_k$ represent the function values and their derivative values at certain node points. This allows them to be used as a parameterization for differentiable functions.

These functions are described in a piecewise sense such that each dimension is partitioned into line segments and the interpolant is a cubic polynomial within these segments. Moreover, the value of the function as well as its derivative is well defined at the nodes. This is achieved by considering the following interpolation for any (arbitrary) interval $x \in [x_0, x_1]$ with $x_0$ and $x_1$ representing the nodes of the segment (element).

$$f(x) = \sum_{i=1}^4 \theta_i h_i^e(\widehat{x}), \qquad \widehat{x} = (x - x_0)/(x_1 - x_0) \tag{34}$$

where,

$$h_1^e = 1 - 3\widehat{x}^2 + 2\widehat{x}^3$$
$$h_2^e = (\widehat{x} - 2\widehat{x}^2 + \widehat{x}^3)(x_1 - x_0)$$
$$h_3^e = 3\widehat{x}^2 - 2\widehat{x}^3$$
$$h_4^e = (-\widehat{x}^2 + \widehat{x}^3)(x_1 - x_0)$$

Moreover the derivatives of the functions are defined as $f'(x) = \sum_{i=1}^4 \theta_i h_i^{e'}$ where

$$h_1^{e'} = (-6\widehat{x} + 6\widehat{x}^2)/(x_1 - x_0)$$
$$h_2^{e'} = 1 - 4\widehat{x} + 3\widehat{x}^2$$
$$h_3^{e'} = (6\widehat{x} - 6\widehat{x}^2)/(x_1 - x_0)$$
$$h_4^{e'} = -2\widehat{x} + 3\widehat{x}^2.$$

Periodicity in the basis functions is introduced by imposing constraints for function values and the derivatives at the boundaries in the 1-dimensional Hermite cubic interpolation.

## F    EXPERIMENTS

This section provides the details of the synthetic problem, the cell migration problem, and the modified Mountain Car problem from the RL benchmark.

### F.1 DATA GENERATION

We first define a state-action value function $Q_{\boldsymbol{\theta}}(\boldsymbol{s}, \boldsymbol{a})$ using the Hermite basis (details provided in Appendix E.3) in the domain of $[-1, 1]^4$. The parameters $\boldsymbol{\theta}$ are provided in the supplemental materials along with the code.

The transition function $T(\boldsymbol{s}'|\boldsymbol{s}, \boldsymbol{a})$ is acquired by Eq. (6) and Eq. (4):

$$T_{\mathrm{MP}}(\boldsymbol{s}', \boldsymbol{a}'|\boldsymbol{s}, \boldsymbol{a}) = T_{\mathrm{MP}}(\boldsymbol{x}'|\boldsymbol{x}) = \left(\frac{\beta}{4\pi\Delta t}\right)^{d/2} \exp\left(\frac{-\beta||\boldsymbol{x}' - \boldsymbol{x} + \nabla\psi(\boldsymbol{x})\Delta t||^2}{4\Delta t}\right), \quad (6)$$

$$T(\boldsymbol{s}'|\boldsymbol{s}, \boldsymbol{a}) = \int_{\mathcal{A}} T_{\mathrm{MP}}(\boldsymbol{s}', \boldsymbol{a}'|\boldsymbol{s}, \boldsymbol{a}) \, d\boldsymbol{a}'. \quad (4)$$

The expert policy $\pi^*(\boldsymbol{a}|\boldsymbol{s})$ is acquired by Eq. (14):

$$\pi^*(\boldsymbol{a}|\boldsymbol{s}) = \frac{\exp(\beta Q_\pi(\boldsymbol{s}, \boldsymbol{a}))}{\int_{\mathcal{A}} \exp(\beta Q_\pi(\boldsymbol{s}, \hat{\boldsymbol{a}})) \mathrm{d}\hat{\boldsymbol{a}}}. \quad (14)$$

The ground truth reward $R(\boldsymbol{s}, \boldsymbol{a})$ (that the agent's policy maximizes) is acquired by Eq. (16):

$$R(\boldsymbol{s}, \boldsymbol{a}) = Q_\pi(\boldsymbol{s}, \boldsymbol{a}) - \gamma \mathbb{E}_{\boldsymbol{s}' \sim T(\cdot|\boldsymbol{s}, \boldsymbol{a}), \boldsymbol{a}' \sim \pi(\cdot|\boldsymbol{s}')} \left[Q_\pi(\boldsymbol{s}', \boldsymbol{a}')\right]. \quad (16)$$

Then, the probability distribution over time $\mathcal{D} = \{p_t(\boldsymbol{s}, \boldsymbol{a})\}_t$ is calculated by

$$p_0(\boldsymbol{s}, \boldsymbol{a}) = \pi(\boldsymbol{a}|\boldsymbol{s}) p_0(\boldsymbol{s}), \quad (35)$$

$$p_t(\boldsymbol{s}', \boldsymbol{a}') = \int_{\mathcal{S} \times \mathcal{A}} p_{t-1}(\boldsymbol{s}, \boldsymbol{a}) T(\boldsymbol{s}', \boldsymbol{a}'|\boldsymbol{s}, \boldsymbol{a}) d\boldsymbol{a} d\boldsymbol{s}, \quad (36)$$

or can also by

$$p_t(\boldsymbol{s}', \boldsymbol{a}') = \pi(\boldsymbol{a}'|\boldsymbol{s}') \int_{\mathcal{S} \times \mathcal{A}} p_{t-1}(\boldsymbol{s}, \boldsymbol{a}) T(\boldsymbol{s}'|\boldsymbol{s}, \boldsymbol{a}) d\boldsymbol{a} d\boldsymbol{s}. \quad (37)$$

Alternately, we can run Monte-Carlo simulation for trajectories using transition function $T(\boldsymbol{s}'|\boldsymbol{s}, \boldsymbol{a})$ and policy $\pi^*(\boldsymbol{a}|\boldsymbol{s})$, and then estimate the probability density from trajectories.

After obtaining the probability distribution over time $\mathcal{D}_p = \{p_t^{\mathrm{data}}(\boldsymbol{s}, \boldsymbol{a})\}_t$ for $t \in [0, \tau]$, we input it as data to the VSI algorithm (Sec. 3.6) and estimate the corresponding potential function $\psi(\cdot)$ from $\mathcal{D}_p$. Leveraging our Conjecture 3.1, the estimated value function $\hat{Q}(\cdot) = -\psi(\cdot)$, and therefore the transition $\hat{T}(\cdot)$, policy $\hat{\pi}(\cdot)$, reward $\hat{R}(\cdot)$ can be obtained through our framework. We then compare the ground truth functions and estimated functions to evaluate the algorithm's performance.

In the convergence analysis, we vary the mesh resolution from 5 to 17 on each dimension. The complete results are provided in the supplemental materials folder "`convergence_analysis`".

### F.2 MODIFIED OPENAI GYM EXAMPLE

Off-the-shelf RL benchmarks (e.g., OpenAI Gym problems) do not directly fall in the category of FP-constrained MDP because their state-action pairs do not necessarily follow the FP dynamics. In this section, we discuss the procedures of transforming an OpenAI Gym example (e.g., Mountain Car) into a form that follows the FP dynamics and present the results of this modified problem in Fig. 5 and 6.

We first obtain the state-action value function $Q(\cdot)$ of the optimal policy in this Mountain Car problem using a RL algorithm (e.g. DDPG, SAC). We approximate it using the Hermite basis in order to have sufficient expressivity for VSI reference. The approximated state-action value function $\hat{Q}(\cdot)$ is shown in Fig. 5a. The probability density data $\mathcal{D} = \{p_t^{\mathrm{data}}(\boldsymbol{s}, \boldsymbol{a})\}_t$ is then generated as the procedures in Appendix F.1. The VSI estimated value function and reward function using the highest resolution mesh are shown in Fig. 5b and 5d, respectively. The KL divergence between data and simulated density $D_{\mathrm{KL}}(p_t||q_t)$ decreases with time, alluding to the convergence to the same steady-state distribution shown in Fig. 6a. As shown in Fig. 6b, The convergence analysis of the reward function shows the error decrease with the higher mesh resolution.

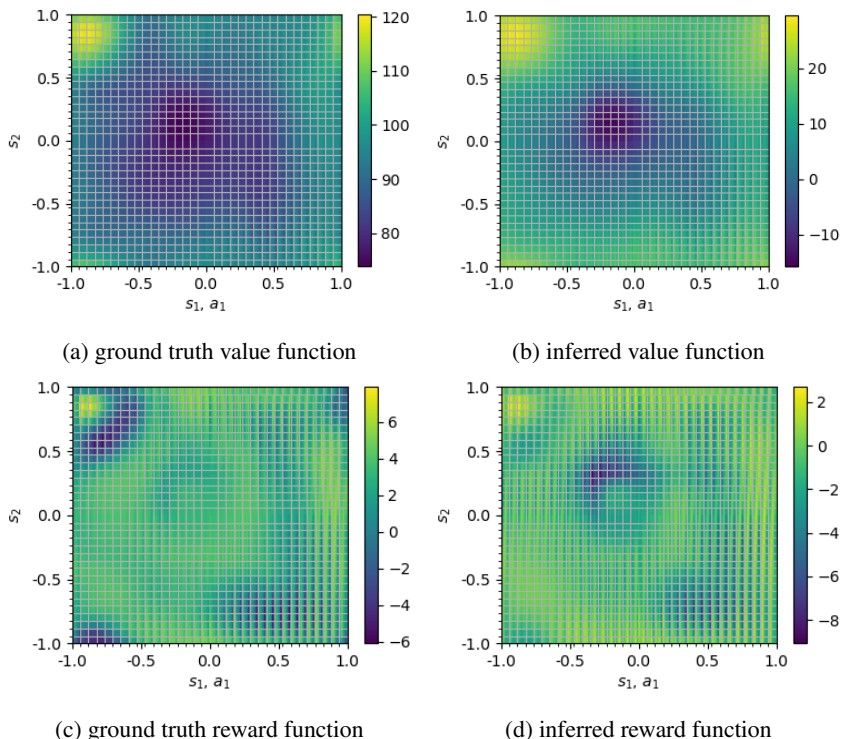

Figure 5: Comparison of inferred value function and reward function (using highest resolution mesh with $N = 34$) with respect to their ground truth.

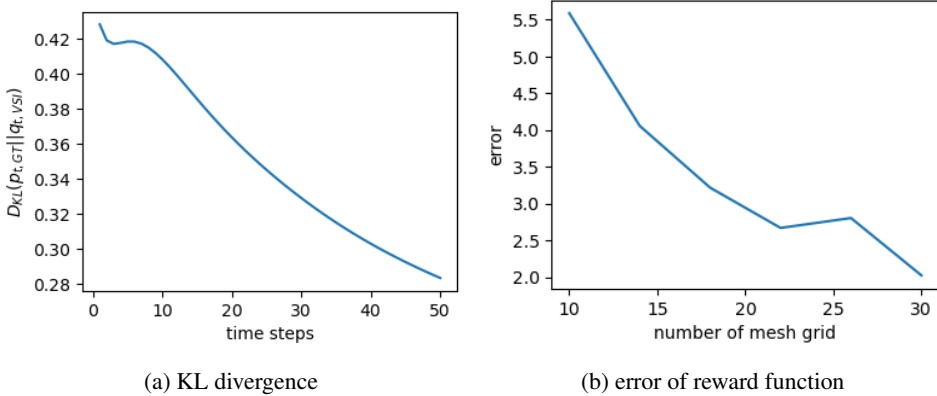

Figure 6: (a) KL divergence $D_{\mathrm{KL}}(p_t || q_t)$ of the probability distribution between data distribution and simulated probability distribution using inferred policy and transition over time. (b) the error of the reward function.

### F.3 EFFECT OF BIAS IN VALUE FUNCTION ESTIMATION

Here, we show that bias in value function (potential function in FP) estimation does not affect the transition and reward derived from our framework.

We denote the ground-truth value function by $Q(\cdot)$, and the estimated value function by $\tilde{Q}(\cdot)$, where $\tilde{Q}(\cdot)$ is shifted by a constant bias $c$ for every state action value: $\forall s \in \mathcal{S}, a \in \mathcal{A}, \tilde{Q}(s, a) = Q(s, a) + c$.

We first show that the bias does not effect the Boltzmann policy:

$$\frac{\exp(\beta\tilde{Q}_\pi(\boldsymbol{s},\boldsymbol{a}))}{\int_{\mathcal{A}}\exp(\beta\tilde{Q}_\pi(\boldsymbol{s},\boldsymbol{a}'))d\boldsymbol{a}'} = \frac{\exp(\beta Q_\pi(\boldsymbol{s},\boldsymbol{a})+\beta c)}{\int_{\mathcal{A}}\exp(\beta Q_\pi(\boldsymbol{s},\boldsymbol{a}')+\beta c)d\boldsymbol{a}'} \tag{38}$$

$$= \frac{\exp(\beta c)\exp(\beta Q_\pi(\boldsymbol{s},\boldsymbol{a}))}{\exp(\beta c)\int_{\mathcal{A}}\exp(\beta Q_\pi(\boldsymbol{s},\boldsymbol{a}'))d\boldsymbol{a}'} \tag{39}$$

$$= \frac{\exp(\beta Q_\pi(\boldsymbol{s},\boldsymbol{a}))}{\int_{\mathcal{A}}\exp(\beta Q_\pi(\boldsymbol{s},\boldsymbol{a}'))d\boldsymbol{a}'}. \tag{40}$$

The transition function is a function of the gradient of the (negative) value function (potential function), and it is trivial to show that the gradient of the value functions with a constant bias are the same:

$$Q(\boldsymbol{s},\boldsymbol{a}) = \tilde{Q}(\boldsymbol{s},\boldsymbol{a}) + c$$
$$\nabla_{\boldsymbol{s},\boldsymbol{a}}Q(\boldsymbol{s},\boldsymbol{a}) = \nabla_{\boldsymbol{s},\boldsymbol{a}}\tilde{Q}(\boldsymbol{s},\boldsymbol{a})$$

Therefore, the dynamics of the system is invariant with respect to the bias term in value function.

Because inverse Bellman equation Eq. (16) is a function of transition, policy, and value function. The bias in the value function will lead to a biased estimation of reward function.

### F.4 CELL RESULTS DISCUSSION

The FP-IRL algorithm applied to the cancer cell dynamics data set yielded the result that the reward is maximized for cells moving leftward and a bit upward (velocities $\boldsymbol{s} = [v_x, v_y]^\top$ in left upper quadrant) in the direction of the chemoattractant, while simultaneously expressing low levels of Akt and high levels of ERK, as shown in Fig. 3c. Each grid square is divided further into a $4 \times 4$ grid for the action $\boldsymbol{a} = [\text{Akt}, \text{ERK}]^\top$ expression levels along the horizontal and vertical directions, respectively. It can be seen that the reward is maximized for low Akt and high ERK levels combined with $[v_x, v_y]^\top$ directed left and upward.

Our biologist collaborators revealed that in their own studies, this treatment led to slightly enhanced migration toward the chemoattractant. Compare Fig. 7a of "control" or untreated cell trajectories with Fig. 7b of trajectories under Alpelisib (Alpe) treatment which causes Akt inhibition, the trajectories in the rightmost plot are slightly longer on average. Finally, Fig. 7c shows cell trajectories under the action of Trametinib (Tram), a drug that inhibits ERK expression. However, in their experiments, Trametinib was not applied, so ERK activity remained high. However, this information was hidden in the data and its importance was realized only after the FP-IRL finding. Most importantly, it suggests that other hidden effects could be "discovered" by the FP-IRL method with an expansion of the action space to include the expression of other markers.

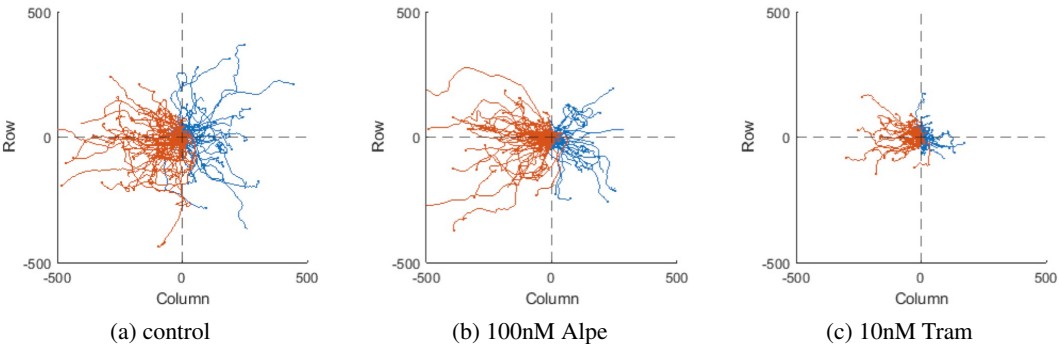

(a) control        (b) 100nM Alpe        (c) 10nM Tram

Figure 7: Centered cell trajectories showing 400 mins to 800 mins of the experiment.

