# OpenReview forum: "FP-IRL: Fokker-Planck-based Inverse Reinforcement Learning --- A Physics-Constrained Approach to Markov Decision Processes"
_ICLR.cc/2024/Conference — Submitted to ICLR 2024_

### Official Review · Reviewer_72Xj · 2023-10-31

**Soundness:** 3 good
**Presentation:** 3 good
**Contribution:** 2 fair
**Rating:** 3
**Confidence:** 4

**Summary:**

The paper proposed to combine physics based dynamics modelling with inverse reinforcement learning. Particularly they consider the class of systems governed by Fokker-Planck equations and derive a IRL method that is computationally efficient. The empirical experiments demonstrate the strength of method on a cancer cell dynamics problem.

**Strengths:**

1. The paper presents a computationally efficient IRL algorithm by leveraging the following properties in FP-dynamics: a. the potential of the system is the negated value function b. The steady state visitation of optimal policy in FP dynamics has a closed form solution in an analytical form of potential function. c. IRL is easy now as given steady state distribution → we can figure out optimal value function and given optimal value function we can extract the reward function.
2. The authors propose to use Hermite cubic functions to induce structure in the potential functions/Q-function that maps state-actions to scalar.
3. Two experiments - 1 on a 2D toy domain and other on a dataset for cancer-cell dynamics show that FP-IRL can recover reward functions and optimal policies well.

**Weaknesses:**

1. I have strong concerns against the motivations in the paper:
    1. “Most IRL methods require the transition function to be known”: This is repeatedly claimed in the paper and is considered as the motivation behind this paper. This is not true as a number of works propose sample-based estimators to solve IRL [1,2,3,4,5,6,7]. These methods should be discussed and compared against.
    2. “Empirical treatment of dynamics using NN’s is not generalizable”: While this statement is true, the way this problem is addressed is not by FP but by using Hermite cubic functions since FP also requires estimating the potential function which has the same learning complexity under a expressible function class. How do other prior methods compare if you use the Hermite cubic function as limited hypothesis class.
    3. “This means that the agent employs to reach states whose value function is high”: This is not a true statement as some high value states can be unreachable under a given starting state distribution. A better explanation for the conjecture might be needed.
2. Novelty: I believe the core strength of paper is in combining FP-dynamics with IRL, merging existing theoretical components. But this combination is not well investigated empirically, experiments are performed on low-dimensional single domain.
3. Empirical evaluation: It is surprising that no baselines were compared against. There are a number of existing IRL baselines[1,2,3,4,5,6,7] that need to be compared in order to make the claims of the paper stand. The function class of Hermite cubic functions should be ablated in these comparisons as well.

[1]: Maximum Entropy Inverse Reinforcement Learning (https://cdn.aaai.org/AAAI/2008/AAAI08-227.pdf)

[2]: Algorithms for inverse reinforcement learning (https://ai.stanford.edu/~ang/papers/icml00-irl.pdf)

[3]: Learning Robust Rewards with Adversarial Inverse Reinforcement Learning (https://arxiv.org/abs/1710.11248)

[4]: IQ-Learn: Inverse soft-Q Learning for Imitation (https://arxiv.org/abs/2106.12142)

[5]: Dual RL: Unification and New Methods for Reinforcement and Imitation Learning (https://arxiv.org/abs/2302.08560)

[6]:OPIRL: Sample Efficient Off-Policy Inverse Reinforcement Learning via Distribution Matching (https://arxiv.org/abs/2109.04307)

 [7]: f-IRL: Inverse Reinforcement Learning via State Marginal Matching (https://arxiv.org/abs/2011.04709)

**Questions:**

None

---

> ### Author Response · Authors · 2023-11-15
> **Reply to Reviewer 72Xj (1/2)**
>
> Thank you for your reviews.
>
> ### Reply to Weakness 1.1:
> - To reiterate our motivation as discussed in the 2nd and 3rd paragraph in Section 1 and last paragraph in Section 3.1:
>     > This is motivated by real-world examples (e.g. cancer cell metastasis), where we can only collect the trajectory data but dynamic knowledge is missing or imperfect, and therefore accessibility of sampling from transition is not available.
> Many systems (e.g., swarms, crowd behavior) have mechanistic foundations in physics, which if exploited can lead to a better understanding and more efficient learning of their incentive structures (i.e., reward function) and transition. With the above motivation, we propose a new method of physics-constrained IRL. This method simultaneously estimates the transition rooted in physics and reward functions using only data on trajectories, while also inferring physical principles that govern the system and using them to constrain the learning.
> - Our method is different than other IRL methods as they would require either the accessibility to sampling from the transition or the analytical form of the transition. Our method does not require both. To reiterate our problem definition: Our target problem setting is that both reward and transition are unknown and need to be inferred in the MDP (i.e., $\mathcal{M} / \{ T(\cdot), R(\cdot) \}$).
> Specifically, the transition is unknown in any of these forms:
>     - The analytical form of the transition function $T(s' | s, a)$ is unknown.
>     - It is not accessible for sampling the next state for a given state and action (i.e., $s' \sim T(s' | s, a)$ cannot be realized). Please note that conventional IRL approaches often assume that sampling is available even though they claim their method does not require the transition (i.e., they do not require the analytical form of the transition but require the ability to sample from the transition.). This may lead to confusion for the audience.
> - We did not conduct an extensive comparison experimentally to other IRL methods because most existing IRL methods require the transition function to be known or **the ability to sample the next state from the transition function**. This is not consistent with our problem setting as these methods do not provide a systematic way to infer the transition or build a model for sampling. It's not directly comparable as the method we propose is designed to simultaneously estimate the transition and rewards under physics constraints.
> - Discussion on given papers: (1) None of these papers assume the transition is not accessible for sampling, and provide a method to infer the transition function as we did. (2) Therefore, they all require at least the sampling accessibility from the transition function in the reward inference (e.g., for collecting and comparing the trajectory of learned policy, or constructing the replay buffer for policy optimization using learned reward). A detailed discussion of each paper is provided below:
>     - Ref. [1]: In lines 2 and 5 of Algorithm 1, the transition is required for the estimation of expected state frequencies of learned policy.
>     - Ref. [2]: In the objective function Eq. (7) on page 4, the analytical form of transition (i.e., transition matrix) is required.
>     - Ref. [3]: In line 4 of Algorithm 1, accessibility to sampling from the transition is required for collecting trajectories and policy optimization (this paper uses TRPO).
>     - Ref. [4]: In Section 5.1, it requires the sampling from the transition for constructing the replay buffer.
>     % Moreover, this method uses IRL for imitation learning. Although it claims it does not require explicit transition knowledge.
>     - Ref. [5]: It requires sampling from the transition either for online or offline settings.
>     % imitation learning, replay buffer
>     - Ref. [6]: In line 1 of Algorithm 1, it requires sampling from the transition to construct the replay buffer.
>     - Ref. [7]: In step 1 in the for-loop of Algorithm 1, it requires sampling from the transition to construct the replay buffer and collecting trajectories of learned policy.

---

> > ### Author Response · Authors · 2023-11-15
> > **Reply to Reviewer 72Xj (2/2)**
> >
> > ### Reply to Weakness 1.2:
> >
> > - As discussed in Sections 1 and 5, we mean that the empirically estimated transition using data alone may not generalize well to the state and action regions away from the data, and may not follow the underlying dynamics.
> > - In this work, processes whose time-continuous form is governed by the FP equation are of fundamental interest. Therefore, by first inferring the governing FP equation (using VSI) and using Conjecture 3.1, we have a transition function that follows the underlying dynamics and has an unambiguous interpretation in terms of physics. The Hermite polynomial basis is only for the convenience of representation and by itself does not grant generalization. This problem indeed is addressed by the FP constraint on the MDP dynamics.
> > - DL and Finite Element approximations offer different ways of parameterizing functions. We choose the Hermite basis functions (a finite element basis) in contrast to DL as it offers a systematic way of approximating differentiable functions. A deep network would not have the issue of discretizing the FP equation by the FE method and bring more scalability in higher dimensional problems. However, the DNN would not allow us to separately extract the potential function from the gradient fields unless a specially designed modular architecture is used for the DNN. Additionally, the expressivity that any chosen DNN depends on the network architecture i.e. to approximate a given function with arbitrary precision, one may require really large networks. This scenario is similar to $h^p$-convergence in mesh-based methods, where higher precision is achieved by reducing the mesh sizes. In our experience, the mesh-based method offers better control in terms of defining the functional spaces used to approximate the value function. Moreover, the tradeoff is that it would be less physically interpretable and more disconnected from the governing physics, and it would require larger trajectory datasets. So under situations where interpretability is less important and data is ample, DL could be an appropriate choice.
> >
> > ### Reply to Weakness 1.3:
> > - In the last sentence of Section 3.3 on page 5, we have the condition that "every state in the MDP is reachable". Our method specializes in problems that follow Fokker Planck dynamics at a continuum scale, so this would not be a limitation.
> >
> > ### Reply to Weakness 2:
> > - VSI can fundamentally handle high-dimensional state-action spaces. However, having its root in finite element methods, this method does suffer from the curse of dimensionality.
> > More precisely, considering a $d$-dimensional state-action space, with a uniform grid of $n$-bins, we get a discrete state-action space with $n^d$ points. The number of operations in the numerical setup of VSI scales linearly with these points i.e. $\sim \mathcal{O}(n^d)$. Therefore, for a $k$-term model for the energy, we require $\sim \mathcal{O}(n^d k)$ operations that result in a linear regression problem with $k$-terms. It should be remarked that the dimensionality of the problem as described here is the dimension of continuous space (total number of states and action variables) and not the cardinality of the discrete space. These operations are vectorized and parallelizable whenever necessary.
> > - Please note that VSI is a published method [Wang et al. (2019; 2021)] that provides a way to infer FP PDE. Our major contribution in this work, as discussed in Section 1, is the FP-IRL algorithm, which inherently does not have the requirement on dimensionality. Future work includes modularizing the FP PDE inference: using other inference methods for higher dimension problems.
> >
> >
> > ### Reply to Weakness 3:
> > - The fundamental difference between our methods and the given papers and the reason for not experimentally existing IRL approaches is discussed in the response to Weakness 1.1.
> > - The reason for choosing the Hermite basis and its role in this work is discussed in the reply to Weakness 1.2.

---

> > > ### Comment · Reviewer_72Xj · 2023-11-18
> > >
> > > **The Hermite polynomial basis is only for the convenience of representation and by itself does not grant generalization. This problem indeed is addressed by the FP constraint on the MDP dynamics.**
> > >
> > > My concern is that this hypothesis is not supported by evidence. For instance, one might be able to show this claim via Theoretical justification or empirically via ablations. What would happen if you replaced Hermite polynomial basis with neural networks? What would happen if you change FP to neural network dynamics?
> > >
> > > Without these comparisons, it's hard to conclude why this method is superior or drives an improvement.

---

> ### Comment · Reviewer_72Xj · 2023-11-18
> **Unclear about role of transitions**
>
> **Our method is different than other IRL methods as they would require either the accessibility to sampling from the transition or the analytical form of the transition. Our method does not require both.**
>
> There are two IRL settings: Offline and Online. Online assuming sampling from the transition distribution by interacting with the simulator or an analytical form.
> Offline does not require either of those. It only requires a fixed dataset of transitions or a fixed dataset of trajectories. As far as I understand that is the setting this paper is operating under the offline setting. I would encourage the authors to provide a more clean and straightforward comparison of their algorithm vs offline IRL algorithms if this is not the case. Algorithm 1 shows that FP-IRL has access to trajectories.
>
> There are a number of IRL algorithms that operate under the offline setting:
> [1]: Chen, Mao, et al. "Batch-Constraint Inverse Reinforcement Learning." PRICAI 2021: Trends in Artificial Intelligence: 18th Pacific Rim International Conference on Artificial Intelligence, PRICAI 2021, Hanoi, Vietnam, November 8–12, 2021,  Proceedings, Part III 18. Springer International Publishing, 2021.
> [2]: (ICLR 23) Luo, Yicheng, et al. "Optimal Transport for Offline Imitation Learning." arXiv preprint arXiv:2303.13971 (2023).
>
>
> Ref. [4] has an offline variant in their paper as well. They also show how rewards can be extracted.
> Ref. [5] above assumes a fixed dataset of transitions for inverse RL as well.
>
>
> Finally, I believe it to be important that the work is properly situated in related prior literature and the discussion of why it is different should be a part of the paper.

---

### Official Review · Reviewer_hceT · 2023-10-31

**Soundness:** 1 poor
**Presentation:** 2 fair
**Contribution:** 2 fair
**Rating:** 3
**Confidence:** 4

**Summary:**

This paper introduces an Inverse Reinforcement Learning (IRL) method that estimates the unknown reward function of a Markov decision process (MDP) without knowing the predefined transition function. Assuming the MDP follows an Ito dynamics, the method infers transitions and reward functions simultaneously from observed trajectories, leveraging mean-field theory through the Fokker-Planck (FP) equation. The authors postulate an isomorphism between time-discrete FP and MDP, which plays the central role in the algorithm design.

**Strengths:**

The strengths of this paper seem unclear as it hinges on a potentially problematic conjecture.

**Weaknesses:**

1. This paper hinges on a conjecture, which might be problematic. This conjecture says that the $\psi$ is equivalent to the $Q$ function in RL problem. However, this $\psi$ function is related to the dynamics. In particular, $\nabla \psi$ is the gradient field. It seems unclear to me why the conjecture is true because the $Q$ function depends on the choice of reward function as well.

It is possible that the authors actually assume that the equivalence is between the (entropy-regularized) optimal policy and the (entropy-regularized) optimal Q function. In this case, it is still unclear to me why the conjecture should be true. It would be great if the authors could at least prove it in special cases such as linear-quadratic case.

2. For discrete-time MDPs, there are existing works that study max-entropy IRL without knowing the transition function. In fact, not knowing the reward seems not a barrier -- we can just estimate the transition from the trajectory data. Thus, the claim that "While most IRL approaches require the transition function to be prescribed or learned a-priori, we present a new IRL method targeting the class MDPs that follow the It^{o} dynamics without this requirement. " seems ungrounded. Moreover, it seems unclear to me why Ito dynamics should be used here.

3. Related work. This work is also related to the huge literature on solving inverse problems involving PDEs, for example, neural ODE and PINN.

**Questions:**

1. Can you prove the conjecture on some toy cases?

---

> ### Author Response · Authors · 2023-11-15
>
> Thank you for your reviews.
> ### Reply to Weakness 1 and Question 1:
> - We agree that there remains theoretical work to be carried out to provide principled guarantees for Conjecture 3.1. There also remains considerable scope for empirical evidence. We hope to work in this direction in future work. For now, we note that this work is motivated by problems of cancer cell dynamics for which our collaborators have generated the real-world example that we used. Cell migration dynamics are widely understood to be governed in the continuous limit by different versions of FP equations. This is the class of problems that this work addresses, by basing itself on Conjecture 3.1. Furthermore, many other real-world problems including Brownian dynamics, swarming and crowd behavior, pattern formation, and morphogenesis are described by FP equations in the continuous limit.  Understanding their incentive structure (i.e., reward function) holds great potential for understanding these complex systems.
> We are not claiming that the conjecture holds uniformly for MDPs. That is likely not the case.
> ### Reply to Weakness 2:
> - Our method is motivated by real-world problems (e.g. cancer cell problems) whose dynamic is known to be governed by physics, specifically that described by the Ito dynamics. A detailed reason for using Ito dynamics can be seen in our motivation as discussed in the 2nd and 3rd paragraph in Section 1 and last paragraph in Section 3.1:
>     > This is motivated by real-world examples (e.g. cancer cell metastasis), where we can only collect the trajectory data but dynamic knowledge is missing or imperfect, and therefore accessibility of sampling from transition is not available.
> Many systems (e.g., swarms, crowd behavior) have mechanistic foundations in physics, which if exploited can lead to a better understanding and more efficient learning of their incentive structures (i.e., reward function) and transition. With the above motivation, we propose a new method of physics-constrained IRL. This method simultaneously estimates the transition rooted in physics and reward functions using only data on trajectories, while also inferring physical principles that govern the system and using them to constrain the learning.
> - Our method is different than other IRL methods as they would require either the accessibility to sampling from the transition or the analytical form of the transition. Our method does not require both. To reiterate our problem definition: Our target problem setting is that both reward and transition are unknown and need to be inferred in the MDP (i.e., $\mathcal{M} / \{ T(\cdot), R(\cdot) \}$).
> Specifically, the transition is unknown in any of these forms:
>     - The analytical form of the transition function $T(s' | s, a)$ is unknown.
>     - It is not accessible for sampling the next state for a given state and action (i.e., $s' \sim T(s' | s, a)$ cannot be realized). Please note that conventional IRL approaches often assume that sampling is available even though they claim their method does not require the transition (i.e., they do not require the analytical form of the transition but require the ability to sample from the transition.). This may lead to confusion for the audience.
> - > not knowing the reward seems not a barrier"
>     - If the reviewer means "not knowing the 'reward' seems not a barrier": Yes, the goal of the IRL problem is to infer the reward function from the trajectory data.
>     - If the reviewer means "not knowing the 'transition' seems not a barrier": there are two ways to infer the transition in the IRL problem:
>        - Inferring the transition before the reward inference. This is what this reviewer means. As we discussed in Section 5 and in 2nd paragraph in Section 2:
>             > "The empirically estimated transition may not inherit the underlying dynamics, and is often challenging to generalize to state and action regions away from training samples relying on data alone."
>
>             > "The lack of interpretability in deep learning models (inferred before the reward inference) can translate to difficulty in scientific understanding of the system behavior.'"
>         - Inferring the transition simultaneously with reward inference. This is what we try to do.
>             > "By constraining with FP dynamics, we systematically reduce this ambiguity to identify a unique pair of transition and reward functions that comply with the FP dynamics."
>
>             This is more beneficial and preferable when the system has a mechanistic foundation and we can exploit it into the inference to reduce the ill-posedness and add physical interpretability.
> ### Reply to Weakness 3:
> - Please note that VSI is a published method [Wang et al. (2019; 2021)] that provides a way to infer FP PDE. Our major contribution in this work is FP-IRL algorithm.
> - PINNS or neural ODE also offers a way to infer the PDE, but they are out of the scope of this IRL work and not included in the related work.

---

> > ### Comment · Reviewer_hceT · 2023-11-22
> >
> > I would like to thank the authors for addressing my concerns.
> >
> > I am still not convinced by the conjecture that $Q = - \psi$ while $\nabla \psi $ is the vector field that drives the dynamics. This assumption is ungrounded because it directly links two seemingly terms that are not directly linked together. In particular, the transition, **together with the reward**, determines the $Q$ function. If the reward is not relevant to $-\psi$, this assumption would definitely fail. It would be great if the authors could verify the conjecture, or find a sufficient condition for it and empirically verify. Because this assumption is fundamental to the proposed method, the method seems also flawed.

---

### Official Review · Reviewer_fsZH · 2023-11-01

**Soundness:** 4 excellent
**Presentation:** 4 excellent
**Contribution:** 4 excellent
**Rating:** 8
**Confidence:** 4

**Summary:**

The manuscript presents a novel method for Inverse Reinforcement Learning (IRL) using a physics-based prior. In essence, the proposed method makes the assumption that underlying dynamics when following the expert policy (induced by the demonstrations) follow Fokker-Planck (FP) equation. With this the authors are able to learn jointly the dynamics and the policy in an elegant way, and perform IRL in systems where the dynamics are not known.

**Strengths:**

- Very cool idea. I really liked it. It is elegant, but although complicated math are involved, I can say that the method itself is simple.
- Well-written paper; although the paper deals with non-trivial and non-popular (in ML) quantities, the authors have made a quite good job in effectively conveying their message.
- Long and interesting discussion section
- Detailed limitations of the method mentioned

**Weaknesses:**

- The evaluation/experiments is quite "weak". There are no baselines and no comparison to state of the art. It is important to know where the new method stands in terms of performance against other state of the art methods with no physics priors.
- No timings are provided for the experiments. Although I tend to agree with the statement "FP-IRL avoids such iterations altogether and instead induces a regression problem leveraging the FP physics that is also computationally more stable", wall-time performance can have very different outcomes (e.g. training the model in the proposed method takes too long).
- More details about the VSI method (in the main text) could help an ML-oriented reader

**Questions:**

- Can you provide some baselines of state of the art methods with no physics priors?
- Can you provide timings of the proposed method?

---

> ### Author Response · Authors · 2023-11-15
>
> Thank you for your reviews.
>
> ### Reply to Weakness 1 and Question 1:
>
> - Our target problem setting is that both reward and transition are unknown and need to be inferred in the MDP (i.e., $\mathcal{M} / \{ T(\cdot), R(\cdot) \}$).
> Specifically, the transition is unknown in any of these forms:
>     - The analytical form of the transition function $T(s' | s, a)$ is unknown.
>     - It is not accessible for sampling the next state for a given state and action (i.e., $s' \sim T(s' | s, a)$ cannot be realized). Please note that conventional IRL approaches often assume that sampling is available even though they claim their method does not require the transition (i.e., they do not require the analytical form of the transition but require the ability to sample from the transition.). This may lead to confusion for the audience.
> - We did not conduct an extensive comparison experimentally to other IRL methods because most existing IRL methods require the transition function to be known or **the ability to sample the next state from the transition function**. This is not consistent with our problem setting as these methods do not provide a systematic way to infer the transition or build a model for sampling. It's not directly comparable as the method we propose is designed to simultaneously estimate the transition and rewards under physics constraints.
>
> ### Reply to Weakness 2 and Question 2:
> - We regret that we did not measure the computation time of our method when we conducted the initial experiment on the cluster, but we reproduced the experiment using lower discretization resolution on the laptop with the Apple M1 Max chip and 32 GB memory:
>
> For the synthetic toy example in Section 4.1 (Please note that original discretization resolution ranges from 5 to 17 in the convergence analysis):
> |     Number of Discretization |            8 |             9 |           10 |           11 |
> |------------------------------|-------------:|--------------:|-------------:|-------------:|
> |     FEA Time (sec)           |        77.05 |         128.6 |       253.56 |       344.04 |
> |     Regression Time (sec)    |         0.77 |          1.12 |         1.64 |         2.46 |
> |     Total VSI Time (sec)         |        77.82 |        129.72 |        255.20 |        346.50 |
>
> For the modified Mountain Car problem in Appendix F.2 (Please note that original discretization resolution ranges from 10 to 30 in the convergence analysis):
> | Number of Discretization |   20  |   25  |   30   |
> |:------------------------:|:-----:|:-----:|:------:|
> | FEA Time (sec)           | 45.95 | 88.54 | 156.29 |
> | Regression Time (sec)    | 0.87  | 1.71  | 3.17   |
> | Total VSI Time (sec)         | 46.82 | 90.25 | 159.46 |
>
> ### Reply to Weakness 3:
> - VSI details are provided in Appendix E and cited papers [Wang et al. (2019; 2021)] as indicated in the second sentence of Section 3.6.
> - Please note that VSI is a published method that provides a way to infer FP PDE. Our major contribution in this work, as discussed in Section 1, is the FP-IRL algorithm.

---

> > ### Comment · Reviewer_fsZH · 2023-11-22
> >
> > Thank you for the comments/replies.
> >
> > > We did not conduct an extensive comparison experimentally to other IRL methods because most existing IRL methods require the transition function to be known or the ability to sample the next state from the transition function
> >
> > This should be stated better or more explicitly and in different places in the manuscript. So that it is super clear to the reader.
> >
> > > Reply to Weakness 3:
> >
> > I agree with the comments, but the ML audience is not very aware of it. Since VSI is needed to understand your method, it'd be easier for the audience to include this in the main text if possible.
> >
> > I have no other comments.

---

### Official Review · Reviewer_GdP4 · 2023-11-02

**Soundness:** 3 good
**Presentation:** 3 good
**Contribution:** 3 good
**Rating:** 3
**Confidence:** 3

**Summary:**

The authors draw an interesting parallel between the Markov decision process (MDP) model commonly used in RL and the Fokker-Planck partial differential equation (FP-PDE) used in modeling physical & biological systems. More specifically, the authors hypothesize equivalence between the $Q$ function in the MDP model and the (negative) potential function $\psi$ in the FP-PDE. This equivalence implies that methods that can be used to learn the potential are valid as IRL methods, using the inverse Bellman equation (IQ-Learn [1]). In this work, the potential is learned using variational system identification (VSI). VSI uses a parameterization scheme where the potential is a product of learnable parameters and features expressed using Hermite cubic functions.The learning occurs through minimization of the magnitude of the FP-PDE functional (since we want the FP-PDE function = 0 eventually). Overall, the established connection is interesting, but the use of VSI is limiting, which is reflected in the experiments.

**Strengths:**

It is known from [2] that the solution of FP-PDE has a Gibbs-Boltzmann density, and in many RL/IRL settings, a Boltzmann policy is assumed, therefore the connection between $\psi$ and $Q$ is easy to grasp. This reveals an intuitive connection to physical phenomena, which is interesting.

**Weaknesses:**

1. Not broadly applicable to more challenging problems, since discretization is required. As the state-action space increases, the approach would become infeasible.
2. No comparison with other IRL methods was provided. For example, is there an advantage to using VSI for the cancer cell problem, over other simple IRL methods?

**Questions:**

1. Page 5: "It imposes a physics constraint that the change in distribution should be small and approach zero over an infinitesimal time step". In the limit $\Delta t \rightarrow 0$, the free energy term should become negligible in Equation 9, right? Why can we then ignore the squared Wasserstein distance term instead of the free energy term?
2. Is VSI parameterization as good as a neural network parameterization, in terms of being able to learn the true function arbitrarily closely?
3. Does Equation 21 have a unique minimizer? If not, how does VSI address the unidentifiability/ill-posedness issue in IRL, i.e. there are many possible rewards (and corresponding value functions) that yield the same policy?
4. (Suggestion) Certain concepts could be introduced more better, through simple examples either in the main text or in the appendix. For example, intution behind FP-PDE - Equation 7, why Wiener processes are used, etc.

**References**

1.  IQ-learn: Inverse soft-Q learning for imitation, Garg et al. (2021)
2.  Free energy and the fokker-planck equation, Jordan et al. (1997)

**Details Of Ethics Concerns:**

_

---

> ### Author Response · Authors · 2023-11-15
> **Reply to Reviewer GdP4 (1/2)**
>
> Thank you for your reviews.
>
> ### Reply to Weakness 1:
> - VSI can fundamentally handle high-dimensional state-action spaces. However, having its root in finite element methods, this method does suffer from the curse of dimensionality. More precisely, considering a $d$-dimensional state-action space, with a uniform grid of $n$-bins, we get a discrete state-action space with $n^d$ points. The number of operations in the numerical setup of VSI scales linearly with these points i.e. $\sim \mathcal{O}(n^d)$. Therefore, for a $k$-term model for the energy, we require $\sim \mathcal{O}(n^d k)$ operations that result in a linear regression problem with $k$-terms. These operations are vectorized and parallelizable whenever necessary.
> - Please note that VSI is a published method [Wang et al. (2019; 2021)] that provides a way to infer FP PDE. Our major contribution in this work, as discussed in Section 1, is the FP-IRL algorithm, which inherently does not have the requirement on dimensionality. Future work includes modularizing the FP PDE inference: using other inference methods for higher dimension problems.
> - As discussed in the **"Applicability"** part in Section 5, there are broad applications in scientific problems:
>     > "With physics-constrained modeling, this allows the application of FP-IRL to problems where the transition is not available for sampling and has not been mathematically modeled, or discovered. Cancer cell migration, as well as the migration of other cell types, is known to be governed by physics, specifically that described by the FP equation (Bressloff, 2014). Therefore, there is interest in the fields of biology, biophysics, and physics more broadly, to have scientific machine learning methods that respect these physics. We achieve this by combining machine learning ideas (IRL) with physics principles (Minimum Energy Principle and FP dynamics). Although the proposed method may not apply directly to some RL problem domains, such as robotics, many other physics phenomena encompassing Brownian dynamics (Keilson and Storer, 1952), swarming (Correll and Hamann, 2015) and crowd behavior (Dogbe, 2010), pattern formation and morphogenesis (Garikipati, 2017) are also described by FP equations in the continuous limit, and this work would also be applicable to them."
> ### Reply to Weakness 2:
> - Our target problem setting is that both reward and transition are unknown and need to be inferred in the MDP (i.e., $\mathcal{M} / \{ T(\cdot), R(\cdot) \}$).
> Specifically, the transition is unknown in any of these forms:
>     - The analytical form of the transition function $T(s' | s, a)$ is unknown.
>     - It is not accessible for sampling the next state for a given state and action (i.e., $s' \sim T(s' | s, a)$ cannot be realized). Please note that conventional IRL approaches often assume that sampling is available even though they claim their method does not require the transition (i.e., they do not require the analytical form of the transition but require the ability to sample from the transition.). This may lead to confusion for the audience.
> - We did not conduct an extensive comparison experimentally to other IRL methods because most existing IRL methods require the transition function to be known or **the ability to sample the next state from the transition function**. This is not consistent with our problem setting as these methods do not provide a systematic way to infer the transition or build a model for sampling. It's not directly comparable as the method we propose is designed to simultaneously estimate the transition and rewards under physics constraints.
> - Our motivation also demonstrates the advantage of FP-IRL in the cancer cell problem as discussed in the 2nd and 3rd paragraph in Section 1 and last paragraph in Section 3.1:
>     > This is motivated by real-world examples (e.g. cancer cell metastasis), where we can only collect the trajectory data but dynamic knowledge is missing or imperfect, and therefore accessibility of sampling from transition is not available.
> Many systems (e.g., swarms, crowd behavior) have mechanistic foundations in physics, which if exploited can lead to a better understanding and more efficient learning of their incentive structures (i.e., reward function) and transition. With the above motivation, we propose a new method of physics-constrained IRL. This method simultaneously estimates the transition rooted in physics and reward functions using only data on trajectories, while also inferring physical principles that govern the system and using them to constrain the learning.
> - A general discussion of the advantages of FP-IRL over other methods is provided in the "Significance" part of Section 5.
> - Again, please note that VSI is a published method that provides a way to infer FP PDE. Our major contribution is the FP-IRL algorithm.

---

> > ### Author Response · Authors · 2023-11-15
> > **Reply to Reviewer GdP4 (2/2)**
> >
> > ### Reply to Question 1:
> >
> > - The free energy $F(p, \psi)$ would not approach zero in an infinitesimal time step, but $\Delta t F(p, \psi)$ could as $\Delta t \rightarrow 0$. However, in an infinitesimal time step, the Wasserstein distance of two distributions $p_t(s, a)$ and $p_{t+1}(s, a)$ would be the dominant term vanishing to zero because it has higher order term on the difference term and change of distribution is small in an infinitesimal time step, and therefore can be neglected in the optimization problem. Then, the $\Delta t$ is a constant coefficient of the free energy and would affect the minimization.
> > - Note that Wasserstein-2 distance between two distributions $p_t(x)$ and $p_{t+1}(x)$:
> >
> > $$ W_2^2(p_t, p) =  \inf_{\gamma \in \Gamma(p_t, p)} \int_{\mathcal{X} \times \mathcal{X}} \gamma(\boldsymbol{x}, \boldsymbol{y}) \left\| \, \boldsymbol{x}-\boldsymbol{y} \right\|_{2}^2 d\boldsymbol{x} d\boldsymbol{y} $$
> >
> > where $\Gamma(p_t, p)$ is the set of coupling of $p_t$ and $p$. A coupling $\gamma$ is a joint probability measure on $\mathcal{X} \times \mathcal{X}$ whose marginals are $p_t $ and $p$ on the first and second factors, respectively.
> >
> > ### Reply to Question 2:
> > - DL and Finite Element approximations offer different ways of parameterizing functions. We choose the Hermite basis functions (a finite element basis) in contrast to DL as it offers a systematic way of approximating differentiable functions. A deep network would not have the issue of discretizing the FP equation by the FE method and bring more scalability in higher dimensional problems. However, the DNN would not allow us to separately extract the potential function from the gradient fields, unless a specially designed modular architecture is used for the DNN. Additionally, the expressivity that any chosen DNN depends on the network architecture i.e. to approximate a given function with arbitrary precision, one may require really large networks. This scenario is similar to $h^p$-convergence in mesh-based methods, where higher precision is achieved by reducing the mesh sizes. In our experience, the mesh-based method offers better control in terms of defining the functional spaces used to approximate the value function. Moreover, the tradeoff is that it would be less physically interpretable and more disconnected from the governing physics, and it would require larger trajectory datasets. So under situations where interpretability is less important and data is ample, DL could be an appropriate choice.
> >
> > ### Reply to Question 3:
> > - Equation 21 does not have a unique minimizer. Again, please note that VSI is a published method [Wang et al. (2019; 2021)] that provides a way to infer FP PDE. IRL ill-posedness is not reduced by VSI.
> > - Instead, IRL ill-posedness is reduced by the physics constraints (i.e., FP dynamic) from our conjecture as we proposed in the paper.
> > - As discussed in Section 5:
> >     > By constraining with FP dynamics, we systematically reduce this ambiguity to identify a unique pair of transition and reward functions that comply with the FP dynamics.
> >
> > ### Reply to Question 4:
> > - Thanks for your suggestion. Introduction and corresponding examples of FP PDE are indeed provided in Section 3.2, for example:
> > - Sentence above Equation 7 explaining FP-PDE:
> >     > "FP-PDE describes the evolution of probability density of states under the Ito SDE."
> > - Sentence above Equation 5 giving examples of Ito SDE and explaining why FP (i.e. Wiener process) is used:
> >     > "Specifically, we target the class of stochastic processes whose dynamics are governed by the Ito SDE (e.g., FP dynamics, many real-world problems including cell dynamics, swarms, and crowd behavior are described by the FP equation as discussed in Sec. 5)"
> > - Sentence under Equation 5 explaining Ito SDE:
> >     > "Thus, the change of state involves directed motion down a potential gradient and diffusion resulting in a random walk from the Wiener process."

---

> > > ### Comment · Reviewer_GdP4 · 2023-11-22
> > > **Rebuttal response**
> > >
> > > > VSI can fundamentally handle high-dimensional state-action spaces.
> > >
> > > From the description, it seems like VSI requires discretization. While VSI is only a part of FP-IRL algorithm, it's a major part of the algorithm since it is required to estimate $\psi$. More experiments on challenging environments with high-dimensional state-action spaces would be needed to establish FP-IRL as a viable alternative to regular IRL methods. These environments do not have to be popular in the RL literature, but they should be more challenging w.r.t. state-action spaces.
> > >
> > > > Instead, IRL ill-posedness is reduced by the physics constraints (i.e., FP dynamic) from our conjecture as we proposed in the paper.
> > >
> > > I think this needs more formal justification.
> > >
> > > Thanks for the clarification regarding the equation with $\Delta t \rightarrow 0$, and for other improvements in terms of clarity. I am inclined to keep my score at this point.

---

### Meta-Review · Area_Chair_cvEk · 2023-12-07

**Metareview:**

In this paper, the authors propose a method to solve the inverse RL (IRL) problem in MDPs with Ito dynamics without knowing or learning the dynamics apriori. The method is based on a conjecture that there exists an isomorphism between discrete-time Fokker-Planck (FP) equation and MDP. This isomorphism allows for inferring the potential function in FP using variational system identification, which consequently allows us to simultaneously evaluate transition and reward functions from observed trajectories.

The work is based on a conjecture, which was not supported by enough evidence to convince the reviewers about its correctness. Moreover, the authors motivate their approach by repeating that "most IRL methods require the transition function to be known", a fact that reviewers found problematic. There are a number of work that use sample-based estimators to solve IRL, which have not been properly discussed in the paper. The empirical results are supposed to provide evidence in support of the conjecture, but they have not been compared with any baseline (any existing IRL algorithm), which the reviewers found problematic. There are statements in the paper that the reviewers found strong and ungrounded. I would suggest that the authors either provide evidence to support them or tone them down.

**Justification For Why Not Higher Score:**

The work is based on a conjecture, which was not supported by enough evidence to convince the reviewers about its correctness. Moreover, the authors motivate their approach by repeating that "most IRL methods require the transition function to be known", a fact that reviewers found problematic. There are a number of work that use sample-based estimators to solve IRL, which have not been properly discussed in the paper. The empirical results are supposed to provide evidence in support of the conjecture, but they have not been compared with any baseline (any existing IRL algorithm), which the reviewers found problematic. There are statements in the paper that the reviewers found strong and ungrounded.

**Justification For Why Not Lower Score:**

None

---

### Decision · Program_Chairs · 2024-01-16

Reject